# Low NO Atmospheric Oxidation Pathways in a Polluted Megacity

Mike J. Newland[1], Daniel J. Bryant[1], Rachel E. Dunmore[1], Thomas J. Bannan[2], W. Joe F. Acton[3], Ben Langford[4], James R. Hopkins[1,5], Freya A. Squires[1], William Dixon[1], William S. Drysdale[1], Peter D. Ivatt[1], Mathew J. Evans[1], Peter M. Edwards[1], Lisa K. Whalley[6,7], Dwayne E. Heard[6,7], Eloise J. Slater[6], Robert Woodward-Massey[8], Chunxiang Ye[8], Archit Mehra[2], Stephen D. Worrall[2,a], Asan Bacak[2], Hugh Coe[2], Carl J. Percival[2,b], C. Nicholas Hewitt[3], James D. Lee[1,5], Tianqu Cui[9], Jason D. Surratt[9], Xinming Wang[10], Alastair C. Lewis[1,5], Andrew R. Rickard[1,5], Jacqueline F. Hamilton[1]

[1]Wolfson Atmospheric Chemistry Laboratories, Department of Chemistry, University of York, York, UK
[2]School of Earth and Environmental Sciences, The University of Manchester, Manchester, UK
[3]Lancaster Environment Centre, Lancaster University, Lancaster, UK
[4]Centre for Ecology and Hydrology, Edinburgh, EH26 0QB, UK
[5]National Centre for Atmospheric Science (NCAS), University of York, York, UK
[6]School of Chemistry, University of Leeds, Leeds, UK
[7]National Centre for Atmospheric Science, School of Chemistry, University of Leeds, UK
[8]Beijing Innovation Center for Engineering Science and Advanced Technology, State Key Joint Laboratory for Environmental Simulation and Pollution Control, Center for Environment and Health, College of Environmental Sciences and Engineering, Peking University, Beijing, 100871, China
[9]Department of Environmental Sciences and Engineering, Gillings School of Global Health, University of North Carolina, Chapel Hill, USA
School of Engineering and Applied Science, Aston University, Birmingham, UK
[10]Guangzhou Institute of Geochemistry, Chinese Academy of Sciences, Guangzhou, China
[a]now at: Chemical Engineering and Applied Chemistry, School of Engineering and Applied Science, Aston University, Birmingham, UK
[b]now at: Jet Propulsion Laboratory, California Institute of Technology, 4800 Oak Grove Drive, Pasadena, CA, USA

*Correspondence to*: Mike J. Newland (mike.newland@york.ac.uk)
Jacqueline F. Hamilton (Jacqui.hamilton@york.ac.uk)

**Abstract.** The impact of volatile organic compound (VOC) emissions to the atmosphere on the production of secondary pollutants, such as ozone and secondary organic aerosol (SOA), is mediated by the concentration of nitric oxide (NO). Polluted urban atmospheres are typically considered to be "high-NO" environments, while remote regions such as rainforests, with minimal anthropogenic influences, are considered to be "low-NO". However, our observations from central Beijing show that this simplistic separation of regimes is flawed. Despite being in one of the largest megacities in the world, we observe formation of gas and aerosol phase oxidation products usually associated with low-NO 'rainforest-like'

atmospheric oxidation pathways during the afternoon, caused by extreme suppression of NO concentrations in the afternoon. Box model calculations suggest that during the morning high-NO chemistry predominates (95%) but in the afternoon low-NO chemistry plays a greater role (30%). Current emissions inventories are applied in the GEOS-Chem model which shows that such models, when run at the regional scale, fail to accurately predict such an extreme diurnal cycle in the NO concentration. With increasing global emphasis on reducing air pollution, it is crucial for the modelling tools used to develop urban air quality policy to be able to accurately represent such extreme diurnal variations in NO to accurately predict the formation of pollutants such as SOA and ozone.

## 1 Introduction

The atmosphere in polluted urban areas has a markedly different chemical composition to that in remote regions (e.g. rainforests). This can lead to changes in the chemical oxidation pathways for volatile organic compounds (VOCs), giving rise to the formation of different secondary pollutants. Oxidation by hydroxyl radicals (OH) is the dominant daytime sink for VOCs, leading to the formation of highly reactive peroxy radicals ($RO_2$). In atmospheres with high concentrations of nitric oxide (NO), emitted by combustion sources such as vehicles, cooking, and energy generation, $RO_2$ radicals react predominantly with NO (Orlando and Tyndall, 2012). This tends to break the initial VOC down to smaller, more oxidised VOCs, and can also produce organic nitrates ($RONO_2$). This pathway also produces $NO_2$, the photolysis of which leads to ozone production. In contrast, in low-NO atmospheres $RO_2$ predominantly react with other $RO_2$, including hydroperoxyl radicals ($HO_2$), or can isomerize/auto-oxidise to form different multi-functionalized oxygenated $RO_2$ (Crounse et al., 2013). These low NO pathways tend to maintain the original carbon skeleton. The large highly oxidised molecules formed can efficiently partition to the aerosol phase to yield secondary organic aerosol (SOA) (Bianchi et al., 2019), which often comprises a large fraction of submicron atmospheric particulate matter (PM) in many regions (Jimenez et al., 2009).

In the past twenty years, emissions, and hence atmospheric concentrations, of nitrogen oxides ($NO_x$) have decreased in urban areas throughout the majority of the developed world. In urban areas this has been due to improvements in vehicle emissions technologies, changes to residential heating, and in many major

European cities, due to restrictions on the types of vehicles that are allowed in certain areas at certain times of the day. In China, through the introduction of the "Air Pollution Prevention and Control Action Plan" in 2013 (Zhang et al. 2019) there has been a concerted effort to reduce pollutant emissions. Numerous pollution control measures have been introduced, including: improved industrial emissions standards; the promotion of clean fuels instead of coal within the residential sector; improving vehicle emissions standards; and taking older vehicles off the road. In Beijing, 900,000 households have converted from using coal to cleaner technologies such as gas or electricity since 2013. These actions have led to a 32 % decrease in $NO_2$ emissions since 2012 (Liu et al., 2016; Krotkov et al., 2016; Miyazaki et al., 2017). Most significant for $NO_x$ emissions however is the stringent vehicle control measures introduced within the last decade, accounting for 47 % of the total reduction in emissions for the city (Cheng et al. 2019). Such reductions in $NO_x$ emissions are expected to lead to an increased importance of low-NO oxidation pathways for VOCs in urban and suburban areas (e.g. Praske et al., 2018). This will lead to the production of a range of low volatility multi-functionalised products, efficient at producing SOA, which have previously been found only in remote environments removed from anthropogenic influence.

Surface ozone in Beijing has increased through the 1990s and 2000s (Tang et al., 2009). The city regularly experiences daily peaks in the summer-time of over 100 ppb (e.g. Wang et al., 2015). Such high ozone episodes are a function both of chemistry and meteorology, with air masses coming from the mountainous regions to the northwest tending to bring in clean air low in ozone, while air masses coming from the densely populated regions to the south and west bring processed polluted air high in ozone (Wang et al., 2017). A number of modelling studies have concluded that the sources of the ozone during high ozone episodes are a combination of both local production and regional transport (Wang et al., 2017; Liu et al., 2019).

Biogenic sources dominate global emissions of VOCs to the atmosphere, with the highly reactive VOC isoprene (2-methyl-1,3-butadiene) contributing ~70% by mass (Sindelarova et al., 2014). The gas and aerosol phase products of isoprene oxidation have been extensively characterized in the laboratory

(Wennberg et al., 2018, and references therein). For isoprene, the low-NO oxidation pathway leads to low volatility products, such as isoprene hydroperoxides (ISOPOOH), that can go on to form significant quantities of SOA via formation of isoprene epoxides (IEPOX) (Figure 1) (Paulot et al., 2009; Surratt et al., 2010; Lin et al., 2012). The high-NO pathway can also form SOA via the formation of methacrolein (MACR), which can react further to form SOA constituents such as 2-methylglyceric acid (2-MGA) and corresponding oligomers (Kroll et al., 2006; Surratt et al., 2006; 2010; Nguyen et al., 2015) (Figure 1). Other significant contributors to isoprene-SOA formed via the high NO pathway include nitrates (e.g. ISOPONO$_2$) and dinitrates (Schwantes et al., 2019). In this work, a suite of isoprene oxidation products, in both the gas and particle phases, are used as tracers of the changing atmospheric chemical environment through the daytime in Beijing.

## 2 Methods

The site was located at the Institute of Atmospheric Physics, between the 3$^{rd}$ and 4$^{th}$ ring road. Measurements took place between 17/05/2017 and 24/06/2017. The site is typical of central Beijing, surrounded by residential and commercial properties and is near several busy roads. It is also close to several green spaces, including a tree-lined canal to the south and the Olympic forest park to the north-east. Isoprene mixing ratios were measured by dual channel gas chromatography (DC-GC-FID). IEPOX/ISOPOOH were observed using iodide chemical ionisation mass spectrometry. The sum of MACR + MVK (m/z 71.05) was measured using proton transfer mass spectrometry. Particle samples were collected onto filter papers at either 3 hourly or 1 hourly time periods, depending on pollution levels. Filters were extracted and analysed with a high throughput method using ultra high-pressure liquid chromatography coupled to a Q-Exactive Orbitrap mass spectrometer. Nitric oxide, NO, was measured by chemiluminescence with a Thermo Scientific Model 42i NO$_x$ analyser. Nitrogen dioxide, NO$_2$, was measured using a Teledyne Model T500U Cavity Attenuated Phase Shift (CAPS) spectrometer. Ozone, O$_3$, was measured using a Thermo Scientific Model 49i UV photometer.

### 2.1 Instrumentation

**DC-GC-FID**

Observations of VOCs were made using a dual-channel GC with flame ionisation detectors. Air was sampled at 30 L min$^{-1}$ at a height of 5m, through a stainless-steel manifold (½" internal diameter). 500 mL subsamples were taken, dried using a glass condensation finger held at -40$^o$C and then pre-concentrated using a Markes Unity2 pre-concentrator on a multi-bed Ozone Precursor adsorbent trap (Markes International Ltd). These samples were then transferred to the GC oven for analysis following

methods described by Hopkins et al (2011).

**CIMS**

A time of fight chemical ionisation mass spectrometer (ToF-CIMS) (Lee et al., 2014; Priestley et al., 2018) using an iodide ionisation system was couple deployed. Experimental set up of the University of

Manchester ToF-CIMS has been previously described in Zhou et al. (2019). During the campaign, gas phase backgrounds were established by regularly overflowing the inlet with dry $N_2$ for 5 continuous minutes every 45 minutes and were applied consecutively. The overflowing of dry $N_2$ will have a small effect on the sensitivity of the instrument to those compounds whose detection is water dependent. Here we find that due to the very low instrumental background for $C_5H_{10}O_3$ and $C_5H_9NO_4$, the absolute error

remains small from this effect (<10 ppt in both reported measurements).

Field calibrations were regularly carried out using known concentration formic acid gas mixtures made in a custom-made gas phase manifold. A range of other species were calibrated for after the campaign, and relative calibration factors were derived using the measured formic acid sensitivity during these calibrations, as has been performed previously (Le Breton et al. 2018, Bannan et al. 2015). In addition to

this, offline calibrations, prior to and after the field work project, of a wide range of organic acids, $HNO_3$ and $Cl_2$ were performed to assess possible large scale sensitivity changes over the measurement period. No significant changes were observed. Offline calibrations after the field work campaign were performed specific to the isoprene oxidation species observed here. IEPOX ($C_5H_{10}O_3$) synthesized by the University of North Carolina, Department of Environmental Sciences & Engineering, was specifically calibrated for.

Aliquots of known concentrations of IEPOX ($C_5H_{10}O_3$) were thermally desorbed into a known continuous flow of nitrogen. For $C_5H_9NO_4$ there was no direct calibration source available and concentrations using

the calibration factor of $C_5H_{10}O_3$ are presented here. Absolute measurement uncertainties are estimated at 50% for the presented IEPOX+ISOPOOH and ISOPONO2 ($C_5H_9NO_4$) signals.

**PTR-MS**

A Proton Transfer Reaction-Time of Flight-Mass Spectrometer (PTR-ToF-MS 2000, Ionicon Analytik GmBH, Innsbruck) was deployed at the base of the 325m meteorological tower at the IAP field site. This instrument has been described in detail by Jordan et al. (2009). The PTR-ToF-MS was operated at a measurement frequency of 5 Hz and an E/N ratio (where E represents the electric field strength and N the buffer gas density) in the drift tube of 130 Td. To enable accurate calibration of the mass scale trichlorobenzene was introduced by diffusion into the inlet stream.

The instrument was switched between two inlet systems in an hourly cycle. For the first 20 minutes of each hour the PTR-MS sampled from a gradient switching manifold, and for the next 40 minutes the instrument subsampled a common flux inlet line running from the 102m platform on the tower to the container in which the PTR-ToF-MS was housed. Gradient measurements were made from 3, 15, 32, 64 and 102 m with air sampled down 0.25 inch O.D. PFA lines and split between a 3 L min$^{-1}$ bypass and 300 ml min$^{-1}$ sample drawn to a 10 L stainless steel container. During the gradient sampling period, the PTR-ToF-MS subsampled for 2 minutes from each container giving an hourly average concentration at each height. In this work, only data from the 3m gradient height is discussed.

Zero air was generated using a platinum catalyst heated to 260 °C and was sampled hourly in the gradient switching cycle. During the field campaign, the instrument was calibrated twice weekly using a 15 component 1 ppmv VOC standard (National Physical Laboratory, Teddington). The calibration gas flow was dynamically diluted into zero air to give a six-point calibration. The sensitivity for each mass was then calculated using a transmission curve. The maximum relative error for PTR-MS calibration using a relative transmission curve has been estimated to be 21% (Taipale et al., 2008). Data was analysed using PTR-MS Viewer 3.

**PM$_{2.5}$ filter sampling and analysis**

PM$_{2.5}$ filter samples were collected using an ECOTECH HiVol 3000 (Ecotech, Australia) high volume air sampler with a selective PM$_{2.5}$ inlet, with a flow rate of 1.33 m$^3$ min$^{-1}$. Filters were baked at 500 $^o$C for five hours before use. After collection, samples were wrapped in foil and then stored at -20 $^o$C and shipped to the laboratory. Samples were collected at a height of 8 m, on top of a building in the IAP complex. Hourly samples were taken on 11th June between 08:00 and 17:00, with one further sample taken overnight. The extraction of the organic aerosol from the filter samples was based on the method of Hamilton et al. (2008). Initially, roughly an 8$^{th}$ of the filter was cut up into 1 cm$^2$ pieces. 4 ml of LC-MS grade H$_2$O was then added to the sample and it was left for two hours. The samples were then sonicated for 30 minutes. Using a 2 ml syringe, the water extract was then pushed through a 0.22$\mu$m filter (Millipore) into another sample vial. An additional 1 mL of water was added to the filter sample, then extracted through the filter, to give a combined aqueous extract. This extract was then reduced to dryness using a vacuum solvent evaporator (Biotage, Sweden). The dry sample was then reconstituted in 1 mL 50:50 MeOH:H$_2$O solution, ready for analysis.

The extracted filter samples and standards were analysed using UPLC-MS$^2$, using an Ultimate 3000 UPLC (Thermo Scientific, USA) coupled to a Q-exactive Orbitrap MS (Thermo Fisher Scientific, USA) with a heated electrospray ionisation (HESI). The UPLC method uses a reverse phase 5 $\mu$m, 4.6 x 100mm, Accucore column (Thermo scientifc, UK) held at 40 $^o$C. The mobile phase consists of LC-MS grade water and 100 % MeOH (Fisher Chemical, USA). The water was acidified using 0.1 % formic acid to improve peak resolution. The injection volume was 2 $\mu$l. The solvent gradient was held for a minute at 90:10 H$_2$O:MeOH, the gradient then changed linearly to 10:90 H$_2$O:MeOH over 9 minutes, it was then held for 2 minutes at this gradient before returning to 90:10 H$_2$O:MeOH over 2 minutes and then held at 90:10 for the remaining 2 minutes, with a flow rate of 300 $\mu$L min$^{-1}$. The mass spectrometer was operated in negative mode using full scan MS$^2$. The electrospray voltage was 4.00 kV, with capillary and auxiliary gas temperatures of 320 $^o$C. The scan range was set between 50 - 750 m/z. Organosulfates were quantified using an authentic standard of 2-MGA-OS obtained from J. Surratt using the method described in Bryant et al. (2019).

**OH measurements**

The OH radical measurements were made from the roof of the University of Leeds FAGE instrument container at the IAP field site. Two Fluorescence Assay by Gas Expansion (FAGE) detection cells were housed in a weather-proof enclosure at a sampling height of approximately 4 m. OH and $HO_2$ radicals were detected sequentially in the first cell (the $HO_x$ cell), whilst $HO_2^*$ and total $RO_2$ radical observations were made using the second FAGE cell (the $RO_x$ cell), which was coupled with a flow reactor to facilitate $RO_2$ detection (Whalley et al., 2018). A Nd:YAG pumped Ti:Sapphire laser was used to generate 5 kHz pulsed tunable UV light at 308 nm and used to excite OH via the Q1(1) transition of the $\mathbf{A^2\Sigma^+, v' = 0 \leftarrow X^2\Pi_i, v'' = 0}$ band. On-resonance fluorescence was detected using a gated micro-channel plate photomultiplier and photon counting. A background signal from laser and solar scatter and detector noise was determined by scanning the laser wavelength away from the OH transition (OHWAVE-BKD). For the entire campaign the $HO_x$ cell was equipped with an inlet pre injector (IPI) which chemically scavenged ambient OH by periodically injecting propane into the air stream just above the FAGE inlet. The removal of ambient OH by chemical reaction provided an alternative means to determine the background signal ($OH_{CHEM-BKD}$) without the need to tune the laser wavelength. By comparison with $OH_{WAVE-BKD}$, $OH_{CHEM-BKD}$ was used to identify if any OH was generated internally within the FAGE cell, acting as an interference signal. In general, good agreement between $OH_{CHEM-BKD}$ and $OH_{WAVE-BKD}$ was observed, with a ratio of 1.07 for the whole campaign (Woodward-Massey, 2018). In this paper, the $OH_{CHEM}$ observations are used. The instrument was calibrated every few days by over-flowing the detection cell inlet with a turbulent flow of high purity humid air containing a known concentration of OH (and $HO_2$) radicals generated by photolysing a known concentration of $H_2O$ vapour at 185 nm. The product of the photon flux at 185 nm and the time spent in the photolysis region was measured before and after the campaign using $N_2O$ actinometry (Commane et al., 2010).

**OH reactivity measurements**

OH reactivity measurements were made using a laser flash photolysis pump-probe technique (Stone et al., 2016). Ambient air, sampled from the roof of the FAGE container, was drawn into a reaction cell at a flow rate of 15 SLM. A 1 SLM flow of high purity, humidified air which had passed by a Hg lamp, generating ~50 ppbv of ozone, was mixed with the ambient air at the entrance to the reaction cell. The

ozone present was photolysed by 266 nm laser light at a pulse repetition frequency of 1 Hz along the central axis of the reaction cell, leading to the generation of a uniform profile of OH radicals following the reaction of $O(^1D)$ with $H_2O$ vapour. The decay in the OH radical concentration by reaction with species present in the ambient air was monitored by sampling a portion of the air into a FAGE cell positioned at the end of the reaction cell. A fraction of the 5 kHz, 308 nm radiation generated by the Ti:Sapphire laser passed through the OH reactivity FAGE cell perpendicular to the air stream, electronically exciting the OH radicals, and the subsequent laser-induced fluorescence signal was detected with a gated channel photomultiplier tube. The 1 Hz OH decay profiles were integrated for 5 minutes and fitted to a first-order rate equation to determine the observed loss rate of OH ($k_{obs}$). The total OH reactivity, k(OH), was calculated by subtracting the rate coefficient associated with physical losses of OH ($k_{phys}$) from $k_{obs}$. $k_{phys}$ was determined by monitoring the decay of OH when the ambient air was replaced with a flow of 15 SLM high purity air. A small correction to account for dilution of the ambient air by the 1 SLM flow of ozone-containing synthetic air was also applied.

## 2.2 Box Modelling

The box modelling that feeds into Figure 3 was performed using the Dynamically Simple Model of Atmospheric Chemical Complexity (DSMACC), zero-dimensional box model (Emmerson and Evans, 2009), together with the isoprene scheme, together with the relevant inorganic chemistry, from the near explicit chemical mechanism the Master Chemical Mechanism (MCM) v3.3.1 (Jenkin et al., 1997; Jenkin et al., 2015). The complete isoprene degradation mechanism in MCM v3.3.1 consists of 1926 reactions of 602 closed shell and free radical species, which treat the chemistry initiated by reaction with OH radicals, $NO_3$ radicals and ozone. It contains much of the isoprene $HO_X$ recycling chemistry identified as important in recent years under "low NO" conditions, including the peroxy radical 1,4 and 1,6 H-shift chemistry described in the LIM1 mechanism (Peeters et al., 2009; 2014), as summarized in Wennberg et al. (2018). Model photolysis rates were calculated using the Tropospheric Ultraviolet and Visible Radiation Model (TUV v5.2) (Madronich, 1993).

The box model was initialised with a range of different concentrations of isoprene (1.7 ppb, 3.4 ppb, 5.0 ppb, 6.7 ppb), and OH (0.25, 0.5, 1.0, 3.0, 10, 20 $\times 10^6$ $cm^{-3}$). [$CH_4$] was fixed at 1.85 ppmv and [CO] at

110 ppbv, T = 298 K, and $[H_2O] = 2.55 \times 10^{17}$ cm$^{-3}$. Entrainment loss rates for all model species were set to $1 \times 10^{-5}$ cm$^{-3}$ s$^{-1}$. For the box model, a column average value for deposition velocity, $V_d$, was calculated according to the functionalities of each species (Table S2). These terms prevent the build-up of secondary products. The values are based on reported deposition rates in Nguyen et al. (2015). A boundary layer height (*BLH*) of 1.5 km was assumed. Loss rates ($L_d$) for each species to dry deposition are then $L_d = V_d/BLH$. Photolysis rates were fixed to mean rates for the day time period 09:00-17:00 calculated for July 1. The model was then run to steady state for a range of fixed NO mixing ratios from $0 - 16{,}000$ pptv.

## 2.3 GEOS-Chem Modeling

GEOS-Chem version 11-01 (http://wiki.seas.harvard.edu/geos-chem/index.php/GEOS-Chem_v11-01) with the inclusion of the aromatic component of RACM2 (regional atmospheric chemistry mechanism 2) was run nested at 0.25 x 0.3125 degree resolution, with 4 x 5 degree boundary conditions using GEOS-FP meteorology. The NO emissions were added via the default MIX emission inventory, which required a 0.9x multiplier on the total daily emissions to match observations from the APHH summer campaign. The diurnal scale factor was considerably steeper than the default GEOS-Chem NO diurnal, with a day-time scale factor on the order of 1.7x and a 0.25x night-time multiplier. Isoprene emissions calculated by the MEGAN biogenic emissions extension were scaled by 2.5x in the Beijing metropolitan region (Jing-Jin-Ji).

## 3 Results

Beijing is a megacity (population of 21.4 M) with an atmospheric reactive VOC mix with both biogenic and anthropogenic influences (e.g. Li et al., 2020). Mean diurnal cycles of ozone, NO, isoprene, and a range of gas and aerosol phase isoprene oxidation products measured at a city-centre site in summer 2017 (Shi et al., 2019) are shown in Figure 2. Data is filtered to only include 'typical' chemistry days, which are considered to be when the ozone mixing ratio increases through the morning to an afternoon peak of > 70 ppb. Such 'typical' days account for 25 of the total of 34 measurement days. Further details of the data filtering is given in Section S1 of the Supplementary Information. Ozone increases throughout the day to a mid-afternoon peak (Figure 2a), driven by the photolysis of $NO_2$, which is rapidly regenerated

through the reactions of ozone, $RO_2$ and $HO_2$ with NO. The high level of ozone acts to suppress NO concentrations. Such a diurnal cycle is typical of urban environments (Ren et al., 2003; Whalley et al., 2018). However, ozone is so high in Beijing, with mixing ratios regularly >100 ppbv in the afternoon, that on many days NO concentrations fall to < 0.5 ppbv in the afternoon, and on some days to < 0.1 ppbv (see Figure S4&5).

The observed diurnal cycles of 'low-NO' and 'high-NO' isoprene oxidation products (Figure 1) in both the gas and aerosol phases can be explained by the observed diurnal cycle of NO (Figure 2c). The high-NO product isoprene nitrate ($ISOPONO_2$), measured using a Chemical Ionisation Mass Spectrometer (CIMS) – see Methods for further details, is produced through the morning from reaction of isoprene

peroxy radicals (ISOPOO) with NO (Figure 2d). During the afternoon, an increasing fraction of ISOPOO begins to react with $HO_2$ or $RO_2$ as the NO concentration drops. This leads to the observed decrease in $ISOPONO_2$, and an increase in the low-NO products IEPOX + ISOPOOH (also measured by CIMS) through the afternoon (Figure 2e). The profile of the signal at m/z 71.05 is assumed to be dominated by the high-NO products MACR+MVK, measured by Proton Transfer Reaction Time-of-Flight Mass

Spectrometer (PTR-ToF-MS) – see Methods for further details. This signal is very similar to that of $ISOPONO_2$ until about 15:00, when it begins to increase, with a second peak observed at around 17:00 (Figure 2f). This latter peak may be an observational artefact as a result of the conversion of ISOPOOH to either MVK (via 1,2-ISOPOOH - the dominant isomer (Reeves et al., 2020)) or MACR (via 4,3-ISOPOOH) on metal surfaces within the inlet of the PTR instrument (Rivera-Rios et al., 2014). Isoprene

oxidation products can also partition into the particle phase and undergo heterogeneous reactions to form organosulfates, with concentrations driven by a number of additional factors such as particulate sulfate and water vapour concentrations. Isoprene organosulfate tracers, 2-MGA-OS (Figure 2g), and 2-methyltetrol-OS (Figure 2h), and were measured on 11 June, with low concentrations through the morning, increasing during the afternoon to a peak around 15:00-16:00. Both are tracers for non-NO

driven chemistry. While 2-methyltetrol-OS is formed via the $ISOPOO+HO_2$ IEPOX pathway (Paulot et al., 2009; Surratt et al., 2010; Lin et al., 2012), 2-MGA-OS (Lin et al., 2013) is formed from the OH initiated oxidation of MPAN (Kjaergaard et al., 2012; Nguyen et al., 2015), with further oxidation leading

to 2-MGA (Surratt et al., 2010; Chan et al., 2010; Nguyen et al., 2015). MPAN is a product of the OH initiated oxidation of MACR in an environment with a high $NO_2/NO$ ratio. So the observation of 2-MGA-OS formation reflects the observed diurnal NO cycle in Beijing. MACR is formed in the morning through the OH oxidation of isoprene in a high NO environment, followed by OH oxidation of MACR in a high $NO_2/NO$ environment in the afternoon to form MPAN, which reacts further with OH to yield 2-MGA.

The observed temporal profiles of the isoprene tracer products suggest a chemical cycle switching from a high-NO chemical regime in the morning, to a regime with a significant contribution from low-NO chemistry in the afternoon in Beijing. First, isoprene nitrates, formed predominantly during the morning (Figure 2d), are characteristic of high-NO chemistry. Second, isoprene hydroperoxides (ISOPOOH) and epoxydiols (IEPOX) (Figure 2e), formed predominantly during the afternoon, are characteristic of low-NO chemistry, where the reaction of ISOPOO with $HO_2$ dominates over reaction with NO. The formation of highly oxygenated molecules (HOMs), characteristic of $RO_2$ isomerisation and auto-oxidation in low NO environments, has also been observed during the afternoon at this site (Brean et al., 2019). Third, the observation of large amounts of 2-methylglyceric acid (2-MGA-OS) (Figure 2g) in the aerosol is suggestive of both high and low NO chemistry having occurred.

## 4 Box Modelling

The chemical box model DSMACC (Emmerson and Evans, 2009), coupled with the near-explicit oxidation mechanism for isoprene from the Master Chemical Mechanism (MCM v3.3.1) (Jenkin et al., 1997; 2015), was used to assess the sensitivity of the fraction of ISOPOO reacting with NO ($f_{NO}$) to varying NO concentrations and OH reactivities ($\sum k_{OH+VOC}$ [VOC]). The model was run to steady state at a range of different fixed concentrations of [OH], [NO], and [isoprene], using fixed photolysis rates typical of Beijing daytime (see Section Methods and Section S5 in Supplementary Information). Figure 3 shows that, as expected, $f_{NO}$ increases with increasing NO concentration. It also shows that $f_{NO}$ is not a fixed value for a given concentration of NO, but decreases with the increasing reactivity of the system (the x-axis in Figure 3). The reactivity varies as a function of the VOC mixing ratios, the reactivity of the VOCs, and the OH concentration, i.e. [OH] × OH reactivity* (Equation E1). Higher reactivity and higher

OH concentrations both lead to a higher concentration of peroxy radicals ($[HO_2] + \Sigma[RO_2]$), reducing $f_{NO}$. Average measurements of ($[OH] \times$ OH reactivity*) and [NO] for the afternoon (12:00 – 20:00) from a range of different environments are shown in Figure 3 (see also Table S1). The $RO_2$ chemistry in the rural southeastern US and the Borneo rainforest lies in the low NO regime (i.e. $f_{NO}$ < 0.5) for the whole afternoon. In the urban areas of London and New York the chemistry remains in the high NO regime through the whole afternoon. However, in Beijing, the extreme suppression of NO concentrations in the afternoon drives the chemistry from a regime in which > 95 % of the $RO_2$ is reacting with NO during the morning, to one in which less than 70 % is reacting with NO by mid-afternoon. $HO_2$ concentrations were measured by FAGE during the campaign (Whalley et al., 2020). Concentrations peaked in mid-afternoon (i.e. when NO is at its lowest), regularly exceeding $5 \times 10^8$ cm$^{-3}$ and reaching up to $1 \times 10^9$ cm$^{-3}$ on some days. Based on the relative reaction rates of $RO_2$ with NO and $HO_2$, for $[HO_2] = 5 \times 10^8$ cm$^{-3}$ the contribution of low NO pathways to $RO_2$ removal would be expected to be roughly 50 % at [NO] = 100 ppt and 10 % at [NO] = 1 ppb.

**5 Discussion and Conclusions**

Modelling was performed with the global chemical transport model GEOS-Chem, with a nested grid at 0.25 x 0.3125 degree resolution over China (see Methods – GEOS-Chem modelling), to investigate the modelled diurnal cycle of NO for Beijing. The results are compared to the measurements for the entire campaign (note not the filtered measurements presented in Figure 2) (Figure 4). These show that while the model does a good job of recreating the measured ozone and $NO_2$ diurnal profiles for the campaign period, it cannot match the observed diurnal profiles of [NO] or the NO to $NO_2$ ratio, particularly at sub ppb levels typically observed during the afternoon. Thus the model will not capture the formation of low NO products from isoprene and other VOCs in Beijing. The major driver of the low NO concentrations in the model is the high levels of ozone, which titrates out the NO. The mean afternoon ozone peak is under predicted by the model by about 10 %. However, this has little impact on the modelled NO concentrations which the model over predicts by a factor of 3 – 5 through the afternoon. As such, with the chemistry currently in the model there is very little flexibility available to appreciably change this

ratio, i.e. changes to other NO sinks in the model, such as $RO_2$, through changes to VOC emissions, will have little effect on [NO]. The fact that the GEOS-Chem modelling cannot recreate the extremely low NO levels observed in the afternoon suggests that there may be additional sinks for NO beyond those currently included in the chemical mechanism. One explanation may be the occurrence of $RO_2+NO$ oxidation pathways that lead to the formation of a second $RO_2$ before forming a stable species, effectively increasing the efficiency of NO to $NO_2$ conversion per initial oxidation step (e.g. Whalley et al., 2020). Such reactions are expected to be particularly important for larger and more complex VOCs, for which the detailed oxidation processes have been less studied and which are heavily parameterised in global models. Auto-oxidation processes that regenerate OH, leading to the formation of further $RO_2$ have also been proposed previously for the high VOC-low NO (< 1ppbv) conditions seen in Beijing and other cities (Hofzumahaus et al., 2009; Whalley et al., 2018; Tan et al., 2019). Inclusion of such $RO_2$ oxidation processes is one of the main foci of the next generation of atmospheric chemical mechanisms. Another explanation may be the presence of high concentrations of other species (not currently included in the chemical scheme) that can rapidly convert NO to $NO_2$, e.g. halogen oxides.

Similar mixed NO regimes as observed here for Beijing have been observed previously at a suburban site in the Pearl River Delta (Tan et al., 2019), and in the semi-rural south east US (Xiong et al., 2015), albeit with lower morning NO peaks. Such a mixed regime will lead to a range of low volatility multifunctional products (Xiong et al., 2015; Lee et al., 2016) some of which are only accessible through this regime, which can efficiently partition to the particle phase to contribute to SOA. With the rates of $RO_2+NO$, and $RO_2+HO_2$ similar for most peroxy radicals (Orlando and Tyndall, 2012), the chemical regime reported herein is not just relevant to isoprene, but to all VOCs (see a comparison for butane and toluene in the Supplementary Information Figure S11).

Our observations from Beijing challenge the commonly accepted view of polluted urban areas as high-NO atmospheric environments in two ways. First, very high ozone (and other sinks) regularly reduces afternoon NO to < 1 ppbv, and on some days to < 0.1 ppbv. This leads to the formation of 'low-NO' products in the gas and aerosol phase. Second, the level of NO that is required for 'low-NO chemistry' to occur is not a fixed value, but is dependent on the concentration and reactivity of the VOCs present and

the concentration of OH. Hence NO concentrations that represent 'low-NO' conditions in a tropical rainforest, for example, are different to those that represent 'low-NO' conditions in a highly polluted urban environment with elevated VOC/OH reactivity.

Under the conditions observed in Beijing, the production of low-NO SOA and the associated increase in PM is shown to be closely linked to photochemical ozone production. Policies that reduce the afternoon ozone peak might also be expected to reduce the production of these aerosol-phase products. However, such policies must also take account of the complex interactions between $NO_x$, VOCs, ozone, and PM. For example, reducing $NO_x$ emissions can counter-intuitively lead to increases in ozone, as has occurred in other major cities (Air Quality Expert Group, 2009), while a recent modelling study (Li et al., 2019) has suggested that reducing PM has led to increases in ozone in China, although a recent experimental study (Tan et al., 2020) in the North China Plain could not observe this effect. With many existing and developing megacities being located in subtropical regions with high emissions of reactive biogenic VOCs, control of which is very difficult, and with continuing reductions in $NO_x$ emissions, such extreme chemical environments as that observed in Beijing can be expected to proliferate. The failure of regional and global models to accurately replicate this chemical regime has wider implications for the prediction of secondary pollutants and hence for determining policies to control air pollution episodes.

**Data availability**

Data are available at http://catalogue.ceda. ac.uk/uuid/7ed9d8a288814b8b85433b0d3fec0300 (last access: 13 Feb 2020). Specific data are available from the authors on request (jacqui.hamilton@york.ac.uk).

**Author Contributions**

JRH, RED, JFH, WJFA, CNH, BL and XW provided the VOC measurements. FAS, WSD and JDL provided the $NO_x$ and $O_3$ measurements. TJB, AM, SDW, AB, CJP and HC collected and analysed the CIMS data. TQ and JDS provided the organo-sulfate standards. DB, WD and JFH provided the organo-

sulfate aerosol measurements. LKW, DEH, EJS, RW-M and CY provided the OH and $HO_2$ data. MN, PME and ARR provided the MCM box modelling. PDI and MJE provided the GEOS-Chem model run. ACL is the PI of the AIRPRO-Beijing project. MJN, JFH and ARR conceived and wrote the manuscript with input and discussion from all co-authors.

## Competing Interests

The authors declare that they have no conflict of interest.

## Acknowledgements

This project was funded by the Natural Environment Research Council, the Newton Fund and Medical Research Council in the UK, and the National Natural Science Foundation of China (NE/N007190/1, NE/N006917/1). We acknowledge the support from Pingqing Fu, Zifa Wang, Jie Li and Yele Sun from IAP for hosting the APHH-Beijing campaign at IAP. We thank Zongbo Shi, Roy Harrison, Tuan Vu and Bill Bloss from the University of Birmingham, Siyao Yue, Liangfang Wei, Hong Ren, Qiaorong Xie, Wanyu Zhao, Linjie Li, Ping Li, Shengjie Hou, Qingqing Wang from IAP, Kebin He and Xiaoting Cheng from Tsinghua University, and James Allan from the University of Manchester for providing logistic and scientific support for the field campaigns. Peter Ivatt acknowledges funding from NCAS through one of its Air Quality and Human Health studentships. Daniel Bryant, William Dixon, William Drysdale, Freya Squires and Eloise Slater acknowledge the NERC SPHERES Doctoral Training Programme (DTP) for studentships.

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

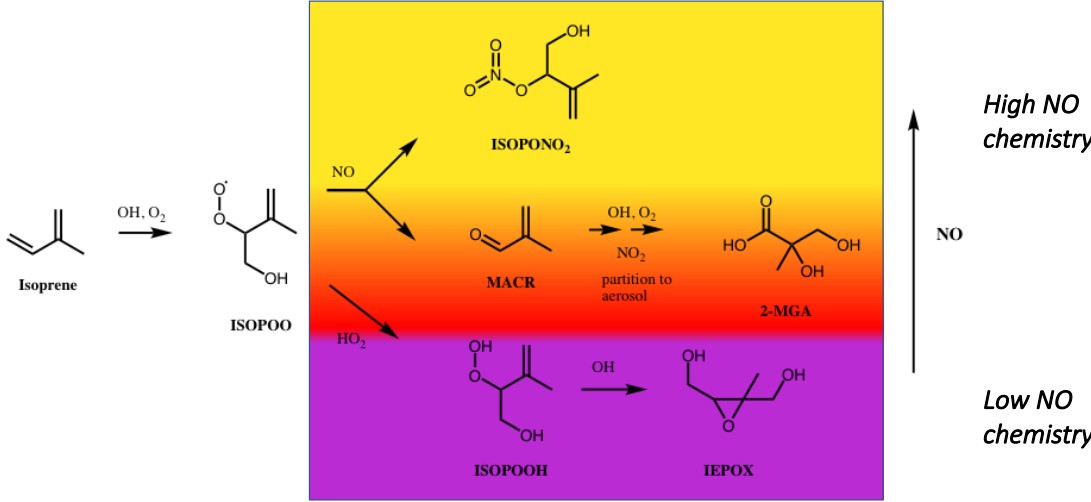

**Figure 1:** Formation pathways of isoprene oxidation products used as tracers of high / low-NO chemistry in this work. Following reaction of the primary VOC, isoprene, with OH, a peroxy radical intermediate (ISOPOO) is formed. At low NO concentrations, ISOPOO reacts with $HO_2$ (or other $RO_2$), to yield hydroperoxide (ISOPOOH) isomers ((4,3)-ISOPOOH isomer is shown), which can be rapidly oxidized to

isoprene epoxydiol (IEPOX) isomers. At high NO concentrations, ISOPOO reacts with NO, a minor product of which is an isoprene nitrate (ISOPONO2). One of the major products of ISOPOO reaction with NO is methacrolein (MACR), the subsequent oxidation of which, in the presence of $NO_2$, can lead to 2-methylglyceric acid (2-MGA) and its corresponding oligomers and organosulfates in the aerosol phase. Measurements of these products in the gas or aerosol phase can be used as tracers for the chemical environment in which they were formed.

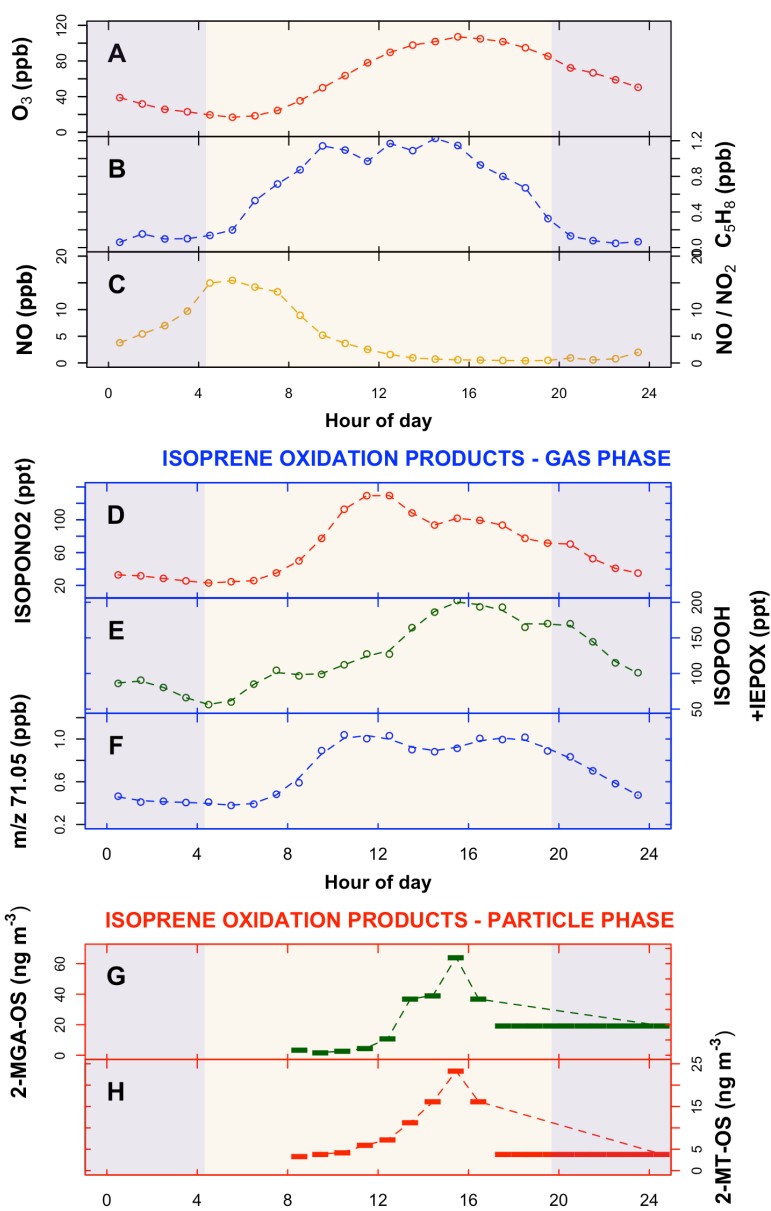

**Figure 2:** Mean diurnal variation of measured organic and inorganic species in the gas phase and aerosol during the Beijing summer observations (data is filtered to only include 'typical' chemistry days – see text for details; the standard deviation of the mean is shown in Figure S3). Blue shaded areas are night-time. a. ozone ($O_3$); (b) isoprene ($C_5H_8$); (c) nitric oxide (NO) and the ratio NO / $NO_2$; (d) the gas phase isoprene 'high NO' oxidation product, isoprene nitrate (ISOPONO2); (e) the isoprene 'low NO' oxidation products ISOPOOH + IEPOX; (f) m/z 71.05, assumed to be predominantly the gas phase isoprene 'high NO' oxidation products methacrolein (MACR) (precursor to 2-MGA) + methyl vinyl ketone (MVK), signal calibrated with MACR/MVK – see text for further details. (g&h) SOA components: 2-methyltetrol-organosulfate (2-MT-OS) and 2-methylglyceric acid-organosulfate (2-MGA-OS), both measured on the 11/12th June 2017, the last filter sample was taken from 17:30 11 June - 08:30 12 June.

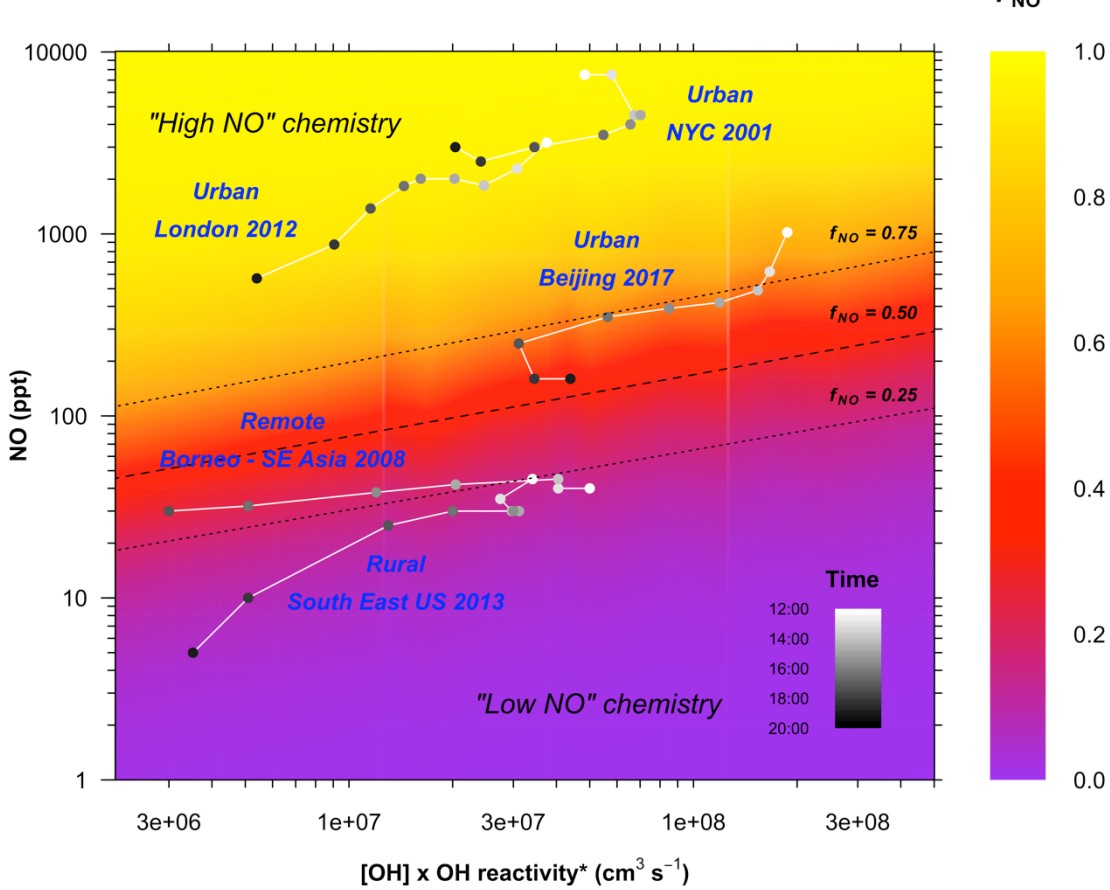

**Figure 3:** Variation of the fraction of ISOPOO reacting with NO as a function of NO concentration and the reactivity of the system. The plot is derived from a series of zero-dimensional box model runs performed as a function of fixed concentrations of [NO], [OH], and [isoprene]. Photolysis is fixed to an average of 09:00-17:00 conditions. OH reactivity* is total OH reactivity of the chemical system minus the contribution from OH + NOx (Equation E1), since these reactions do not produce RO2.

OH reactivity* = $\sum k_{OH+VOC}$ [VOC]    (E1)

The dashed line shows the fraction of ISOPOO reacting with NO $f_{NO} = 0.50$, dotted lines show $f_{NO} = 0.25$ and 0.75. Points are average diurnal hourly measurements of NO, OH, and OH reactivity* for the period 12:00 – 20:00 pm from a range of different environments: The rural sites, Borneo (Whalley et al., 2011) (only shown for 12:00-18:00) and the Southeast US (Sanchez et al., 2018), and the urban sites London (Whalley et al., 2016), New York City (Ren et al., 2003), and Beijing (this work). See the SI for full details.

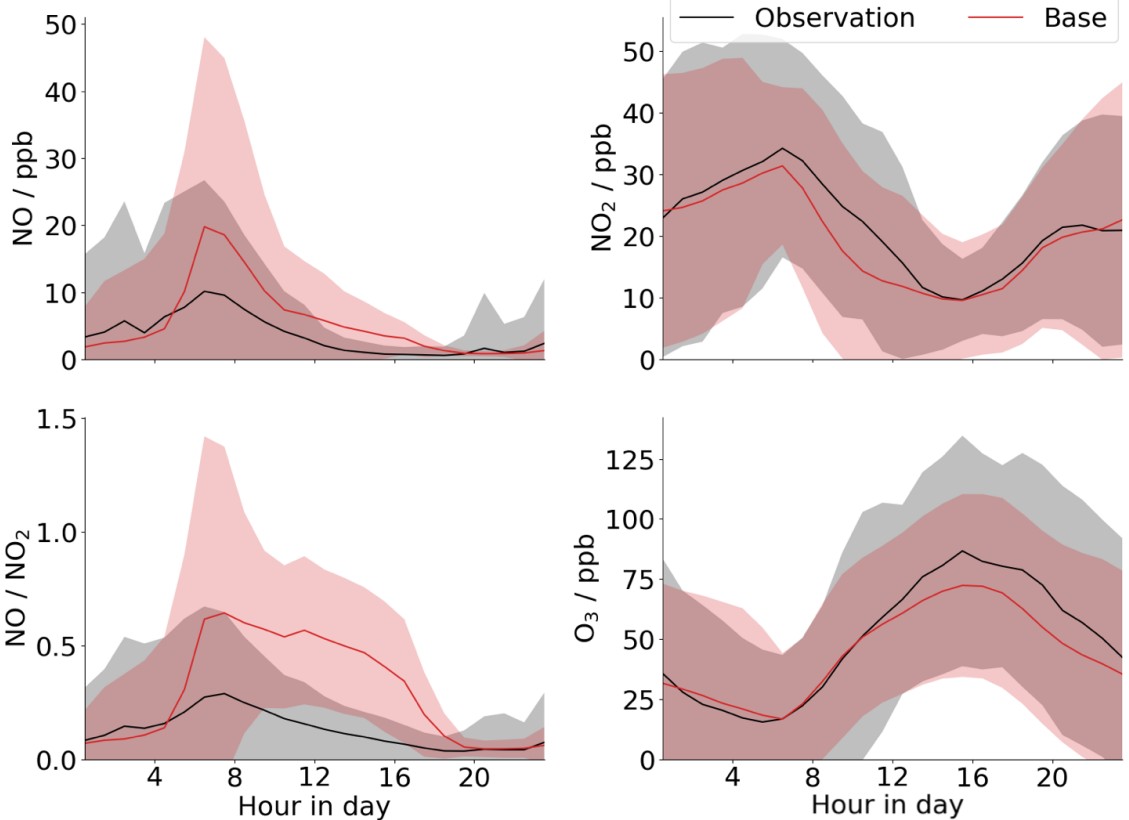

**Figure 4:** Comparison of GEOS-Chem model output (red) to mean diurnal measurements for the entire campaign (black) for NO, NO$_2$, O$_3$
and the NO/NO$_2$ ratio. Shaded regions are two standard deviations of the mean.