# Peer review of "Low NO Atmospheric Oxidation Pathways in a Polluted Megacity"

_Atmospheric Chemistry and Physics, 2020_

## Referee Comment (RC1) · Anonymous Referee #2 · 12 Mar 2020

Overall:

This paper is novel and describes interesting field campaign results in Beijing, China that show through isoprene oxidation tracers that low-NO chemistry is important in the afternoon in Beijing, China. The paper emphasizes the importance that megacities especially those surrounded by vegetation with large biogenic VOC sources and high NOx levels likely have both low- and high-NO chemistry and modeling tools that are used for policy need to represent both of these pathways accurately.

Although the core of the paper is quite exciting and promising, there are gaps in the paper, which require more description before publication. The paper is well-written, but not well-organized. As described below, expanding on several sections in the main text and moving several sections from the supplement into the main text is needed before

final publication.

In general,

The paper in the abstract, conclusions, and throughout, mentions the importance of accurately representing both low- and high-NO chemistry. Most models do represent both of these processes reasonably well at least for isoprene itself. Perhaps, you can expand on what you mean by this further. It may be more important to acknowledge the importance of mixed regimes. The conditions that occur in Beijing in the late afternoon are similar to the regime that occurs in the Southeast U.S. in the afternoon too. In the southeast U.S., isoprene hydroxy nitrates (formed from RO2 + NO channel) react with OH to form peroxy radicals that then react with HO2 to form isoprene dihydroxy hydroperoxy nitrates. These products formed from both RO2 + NO and RO2 + HO2 channels are significantly less studied, but important in regimes like you describe here. See references such as Xiong et al., 2015, Observation of isoprene hydroxynitrates in the southeastern United States and implications for the fate of NOx and Lee et al., 2015, Highly functionalized organic nitrates in the southeast United States: Contribution to secondary organic aerosol and reactive nitrogen budgets.

The format of the paper makes the paper harder to follow and is not similar to what is typically done in ACP. The methods are at the end of the paper instead of in the middle. The methods should be moved after the introduction and before the results. The results section should reference the methods section as needed. Additionally, substantial information is contained in the supplement with only 3 figures in the main text. Redistributing some of the more pertinent information and figures that are currently in the supplement into the main text would be beneficial. In particular and as described below, the section on GEOS-Chem modeling with Figure S10 and the section describing how you filtered the mean diurnal plots should definitely be in the main text.

Additionally, references to the supplement just state "see Supplementary Information". For information that remains in the supplement after addressing the comment above,

please provide more detail either by adding section numbers in the supplement or referring to specific supplementary figures.

Specific comments

Introduction

There are many past papers describing atmospheric chemistry and air pollution in China and Beijing. Some summary of these studies and how this study is similar or different is warranted to put this paper in context. For example, Wang, 2017, Ozone pollution in China: A review of concentrations, meteorological influences, chemical precursors, and effects summarizes many studies.

Page 2 line 66

Please include a reference here that describes the VOC mixture in Beijing. Additionally, you could add the VOC mixture at your sampling location here. How important is isoprene compared to the other VOCs you measure with the DC-GC-FID? How does your VOC mixture compare to other VOC mixtures measured in Beijing or comparable cities in China?

Figure 2

It would be clearer to the reader to overlay b, c, and d on top of one another and expand these figures horizontally to that of figure a. Also to overlay e and f and also expand this one horizontally to figure a. This way the NO concentration and the region of low NO chemistry can be directly seen in all of the figures.

Adding OH, HO2, and NO2 concentration to Figure 2 panel would be beneficial to highlight the low NO/NO2 ratio mentioned throughout the text and demonstrate how OH and HO2 change diurnally.

The Section on "filtering data for mean diurnal plots" in the supplement should either be moved to the main text or summarized in the main text as well as in the Figure 2

caption. Without reading the supplement, the reader would not know that any filtering was done to the data. If possible, please also provide the mean diurnal profile for the same species listed in Figure 2 for the "atypical" days. This way one could contrast how the chemistry differs between a "typical" day where ozone gets above 70 ppb in the afternoon and an "atypical" day, which still occurs 25% of the time, where ozone is lower than 70 ppb. Also provide some discussion on how these products differ on "typical" and "atypical" days.

Page 3 line 84

Please include references for these organosulfate tracers. Also you mention 2-MGA-OS is a tracer for high-NO SOA. Technically it is more of a tracer for high-NO and high-NO2 SOA. On page 3 line 93, you state that 2-MGA-OS is suggestive of both high and low NO chemistry. Please make sure these statements are consistent and include appropriate references.

Page 3 line 91

HOMs are mentioned to have been detected at this site. There are potentially products from mixed regimes that could be detected in the late afternoon. For example, other studies have detected isoprene nitrooxy hydroxy hydroperoxides in the late afternoon when NO concentrations drop in the Southeast U.S. (see explanation above in general comments). Can you detect isoprene nitrooxy hydroxy hydroperoxides with your CIMS instrument? It would add a lot to your paper to add this compound to Figure 2 if you can detect it. This would also help to address the general comment, that it's not only about getting low- and high- NO chemistry correct, but representing products from mixed regimes that are not always incorporated into reduced chemical mechanism used in global and regional models.

Figure 3.

It would be beneficial to the reader to add the year for all the campaigns (not just for

NYC) as the fno will greatly depend on when the measurements were taken and how much NOx was present.

Page 4 line 115

The fact that GEOS-Chem cannot represent this chemistry is important and should not be hidden in the supplement. I recommend moving the description of the GOES-Chem modeling and Figure S10 into the main text or providing significantly more detail here.

GEOS-chem is a global model, so please rephrase "regional chemical transport model" to include a global model nested over China or some such explanation.

Figure S10 needs labels for which red or black lines are model versus observations.

There is recent evidence to suggest that aerosols play an important role in accurately representing ozone in China through loss process of NOx and HOx (Li et al., 2019). In your GEOS-chem simulations, do you assume the same aerosol uptake coefficients as this study, and if not could this impact your results in Figure S10?

Comparing more compounds (OH, HO2, isoprene, other VOCs measured by the GC-FID, ISOPOOH + IEPOX, isoprene hydroxy nitrates) with the GEOS-chem results would make your paper much more significant and help better understand how well models are representing the chemistry you describe in Beijing. Can you add this?

If aerosols are important in China for representing O3-NOx-HOx (and in particular the loss of HO2 to aerosols), how would this impact your results with the box-modeling in Figure 2, which only includes gas-phase chemistry.

On page 5 line 137, Jacob et al., 2019 is referenced, but not incorporated in the reference list. Please update this reference or use Li et al., 2019.

Page 9 line 274.

What was the relative humidity used in the box-model?

---

## Referee Comment (RC2) · Anonymous Referee #1 · 26 Mar 2020

General Comments: The authors summarize their results in the Abstract: "Despite being in one of the largest megacities in the world, we observe significant formation of gas and aerosol phase oxidation products associated with the low-NO 'rainforest-like' regime during the afternoon. This is caused by a surprisingly low concentration of NO, coupled with high concentrations of VOCs and of the atmospheric oxidant hydroxyl (OH). Box model calculations suggest that during the morning high-NO chemistry predominates (95%) but in the afternoon low-NO chemistry plays a greater role (30%)."

In pristine "Rainforest-like" conditions O3 production is NOx limited and OH-reactivity is mainly controlled by isoprene (Wei et al. 2019). O3 in Beijing is largely driven by transport of highly chemically processed air across densely populated areas of 500 million people in the south (your SI, Parrish et al. 2016). Your selection of "typical"

pollution days was made accordingly (your SI). This means that air reaching the measurement point in the afternoon in Beijing contains a large fraction of highly processed VOC originating from more distant urban sources. Many different VOC preferable different alkenes contribute to Ozone formation. The individual contributions most likely will change during the day. Alkyl nitrates are key compounds in controlling tropospheric oxidants and the lifetime of NOx (Teng et al. 2015). During daylight hours alkyl nitrates are produced via radical chain terminating branch reactions from RO2 and NO. The other larger branch recycles HOx and produces O3. In simple terms: increasing the carbon number increases the alkyl nitrate branch. Diurnal variation of individual alkyl nitrates, which should have been measured with the I- CIMS (Lee et al. 2016), will allow to estimate the contribution of individual alkenes (VOCs) to the O3 formation (Teng et al., Fig. 8). Even if quantitative alkyl nitrate sensitivities are not available it would show how important isoprene oxidation is over the course of the day.

While I find the study interesting it does not offer a clear explanation how high afternoon O3 at rather low NO occur. The title is misleading and has to be changed. A suit of instruments was simultaneously analyzing the air composition in Beijing. Only signals focusing on isoprene oxidation are chosen and presented in this manuscript. It is not clear why this selection was made. In any case in the present form, it does not give a conclusive picture of summer time chemistry in Beijing.

Lee et al. PNAS 113 (2016) 1516-1521 Teng et al. Atmos. Chem. Phys., 15 (2015) 4297–4316 Wei et al. Atmos. Environ. 206 (2019) 280-292

Specific comments Fig. 2 depicts mean diurnal variation of measured species during the Beijing summer observations. The authors use Fig. 2 to justify their conclusions of changing chemistry from high NO in the morning to low NO in the afternoon. While NO measurements clearly show that. The offered explanation using the diurnal behavior of isoprene oxidation products are less clear. - High NO and low NO organosulfate tracers 2-MGA-OS (Fig. 2e) and 2-methyltetreol-OS (Fig. 2f) from the particle phase show almost exactly the same diurnal behavior – a pronounced peak at 15:00 (on one

day). - ISOPONO2 concentrations at 15:00 are still above 100 ppt. (Fig.2c) - Fig. 2d shows a double peak behavior and does not help at all. The PTR-MS signal at m/z 71.05 is not MACR+MVK as indicated in the Figure (see methods discussion later) - NO2 and OH is not shown.

-Observed low NO values cannot be explained solely by the increased O3 in the afternoon according to GEOS-Chem. -One explanation would need additional NO sinks that recycle OH without producing O3. -Another speculation would need the presence of halogen oxides.

Methods Native speakers (there are a few co-authors) should help to improve especially the methods part.

CIMS Discuss in more detail how the CIMS was calibrated for the isomers IEPOX and ISOPOOH, respectively. The methods description suggests that only IEPOX standards were available. Discuss measurement errors for your conditions. No calibration standards for ISOPONO2 were available. Xiong et al. 2015 found different sensitivity and stability for different ISOPONO2 isomers using I- CIMS. Discuss how this impacts the quality of your ISOPONO2 data. How are isomer specific inlet line losses estimated and corrected?

Xiong et al. Observation of Isoprene Hydroxynitrates in the Southeastern United States and Implications for the Fate of NOx. Atmos. Chem. Phys. 2015, 15, 11257−11272.

PTR-MS The PTR-MS instrument description suggests that ambient air containing varying ozone concentrations (low in the morning and very high in the afternoon) is sampled through a 10 L stainless steel container. Your description is not detailed enough to gain the "storage time" in this container. Describe which tests were performed to exclude/estimate production of compounds, which are monitored as 71.05 m/z ions from O3 + VOC reactions in the stainless steel container. How log is the storage time? Such artificial reactions could contribute to the observed "second" peak at 16:00 -20:00 which follows the O3 concentration rather than isoprene in Fig. 2d. You

mention correctly that ISOPOOH isomers are converted to MVK and MACR on stainless steel surfaces even at room temperature. If the sample air is stored in a stainless steel container for longer than seconds then the signal at mass 71.05 in Fig. 2d by no means can be assigned to MACR+MVK only! Describe what happens to ISOPOOH and IEPOX in your special inlet design.

---

## Author Comment (AC1) · 22 May 2020

**Response to Reviewers of:**

**Rainforest-like Atmospheric Chemistry in a Polluted Megacity by Newland et al., 2020, submitted to ACP**

**General Response**

We thank the reviewers for giving their time to make insightful comments, helping to clarify and further improve our manuscript. The referees recognise the importance of the results presented, and recommend publication in ACP after some changes.

**Formatting:** Reviewer #2 makes the point that the formatting is not as expected from an ACP Article. Indeed this is correct. However, this is because we have submitted this manuscript with the intention of it being an ACP Letter, and have followed the formatting guidelines for this format (https://www.atmospheric-chemistry-and-physics.net/about/manuscript_types/acp_letters.html). However, there is currently not a mechanism to submit the manuscript as a Letter, with the designation coming after review. Following discussion with the Copernicus editorial team and ACP editors, it was decided that the comments should be addressed with the intention of the manuscript being published as a Letter, and if subsequently it was considered more appropriate for the manuscript to be an Article, then it would be re-formatted as such.

**Overall content and scope:** As a Letter, the aim of this manuscript is a tight focus on the key, high impact results. That is, evidence of dramatically changing oxidation pathways during the daytime in the summer in Beijing, highlighting the impact of this observation on VOC oxidation chemistry (note, not specifically isoprene chemistry, we are using the measurement isoprene oxidation products as photochemical markers of the changing chemical pathways throughout the daytime), and the inability of models to capture this diurnal change in the oxidation pathways. Again, the paper is not specifically about Beijing per se, but as we discuss, is likely to be relevant to many megacities, particularly in the (sub)tropics, with emphasis on reducing NOx emissions but still having a high loading of reactive VOCs. The story told by the measured isoprene oxidation products is corroborated by the auxiliary NOx and ozone measurements.

What the manuscript is not aiming to do is to describe the VOC composition in Beijing, talk about the role of isoprene chemistry in Beijing, or talk about ozone production in Beijing (although, some of this will be the subject of forthcoming publications). We feel that both reviewers have slightly misunderstood this, which is clearly a reflection on how we have written the manuscript. In light of this we have made several changes to the manuscript to try to make these aims clearer. This has begun with the title which we have altered slightly to better highlight the focus of the paper, changing it from: '*Rainforest-like Atmospheric Chemistry in a Polluted Megacity*' to '*Rainforest-like Atmospheric Oxidation Pathways in a Polluted Megacity*'. We have also made changes to the abstract and the introduction. Both reviewers have suggested changes to extend the paper to a more general discussion of the atmospheric chemistry of Beijing, however, as discussed above we do not feel that this is within the scope of the paper, nor necessary background for the scientific points that are being made. What we have done is to add some background on changing NOx concentrations in urban environments, and in particular in Beijing, as well as the causes of high ozone episodes in Beijing. We feel that this bears more relevance to the manuscript, rather than a broad discussion on either the VOC mix, or ozone production – neither of which are a focus of this Letter. All changes to the manuscript are in line with the reviewers' comments and suggestions.

Responses to each reviewer are given below. Responses to specific points raised by each reviewer are given separately beneath that point. Reviewers' comments are bold and italic, the authors' comments are inset in plain type.

**Anonymous Referee #1**

**General Comments**

*The authors summarize their results in the Abstract: "Despite being in one of the largest megacities in the world, we observe significant formation of gas and aerosol phase oxidation products associated with the low-NO 'rainforest-like' regime during the afternoon. This is caused by a surprisingly low concentration of NO, coupled with high concentrations of VOCs and of the atmospheric oxidant hydroxyl (OH). Box model calculations suggest that during the morning high-NO chemistry predominates (95%) but in the afternoon low-NO chemistry plays a greater role (30%)."*

*In pristine "Rainforest-like" conditions O3 production is NOx limited and OH-reactivity is mainly controlled by isoprene (Wei et al. 2019). O3 in Beijing is largely driven by transport of highly chemically processed air across densely populated areas of 500 million people in the south (your SI, Parrish et al. 2016). Your selection of "typical" pollution days was made accordingly (your SI). This means that air reaching the measurement point in the afternoon in Beijing contains a large fraction of highly processed VOC originating from more distant urban sources. Many different VOC preferable different alkenes contribute to Ozone formation. The individual contributions most likely will change during the day. Alkyl nitrates are key compounds in controlling tropospheric oxidants and the lifetime of NOx (Teng et al. 2015). During daylight hours alkyl nitrates are produced via radical chain terminating branch reactions from RO2 and NO. The other larger branch recycles HOx and produces O3. In simple terms: increasing the carbon number increases the alkyl nitrate branch. Diurnal variation of individual alkyl nitrates, which should have been measured with the I-CIMS (Lee et al. 2016), will allow to estimate the contribution of individual alkenes (VOCs) to the O3 formation (Teng et al., Fig. 8). Even if quantitative alkyl nitrate sensitivities are not available it would show how important isoprene oxidation is over the course of the day.*

> We agree that the high ozone observed in Beijing is likely largely driven by regional sources (which subsequently titrates out the NO in Beijing itself in the afternoon). However, in this manuscript we are not seeking to identify the drivers of ozone production in Beijing. We do not state anywhere that isoprene is (or is not) important for ozone production. Isoprene oxidation products are being used as tracers of the chemical environment in which they were formed. This point should now be clearer in the manuscript.

*While I find the study interesting it does not offer a clear explanation how high afternoon O3 at rather low NO occur.*

> Again, this is not the point of the study and we make no attempt to look at the drivers of ozone production.

*The title is misleading and has to be changed.*

We have now changed the title slightly to: '*Rainforest-like Atmospheric Oxidation Pathways in a Polluted Megacity*'. Hopefully this helps to emphasise the point that we make in the abstract, that by 'Rainforest-like' we mean that $RO_2$, specifically ISOPOO, are reacting with $HO_2$ / $RO_2$ rather than with NO as might be expected in an urban environment. We are not referring to the ozone production regime of a rainforest.

*A suit of instruments was simultaneously analyzing the air composition in Beijing. Only signals focusing on isoprene oxidation are chosen and presented in this manuscript. It is not clear why this selection was made. In any case in the present form, it does not give a conclusive picture of summer time chemistry in Beijing.*

*Lee et al. PNAS 113 (2016) 1516-1521 Teng et al. Atmos. Chem. Phys., 15 (2015) 4297–4316 Wei et al. Atmos. Environ. 206 (2019) 280-292*

We make it clear in the abstract and introduction that isoprene oxidation products, of which we have an extensive suite of measurements in both the gas phase and particle phases, are used in order to identify the chemical environment in which they were formed. The conclusions drawn from these measurements are backed up by the measurements of NO, $NO_2$ and $O_3$. This then provides not only a description of the changing diurnal profile of the oxidation state of the local Beijing atmosphere, but also provides convincing arguments that we have a reasonable understanding of the production pathways of these products in the ambient environment, as opposed to solely from laboratory studies. The aim of the paper is not to give a comprehensive picture of summertime chemistry in Beijing – the focus is on how the chemical environment changes through the day.

**Specific comments**

*Fig. 2 depicts mean diurnal variation of measured species during the Beijing summer observations. The authors use Fig. 2 to justify their conclusions of changing chemistry from high NO in the morning to low NO in the afternoon. While NO measurements clearly show that. The offered explanation using the diurnal behavior of isoprene oxidation products are less clear. High NO and low NO organosulfate tracers 2-MGA-OS (Fig. 2e) and 2-methyltetreol-OS (Fig. 2f) from the particle phase show almost exactly the same diurnal behavior – a pronounced peak at 15:00 (on one day). - ISOPONO2 concentrations at 15:00 are still above 100 ppt. (Fig.2c) - Fig. 2d shows a double peak behavior and does not help at all. The PTR-MS signal at m/z 71.05 is not MACR+MVK as indicated in the Figure (see methods discussion later) - NO2 and OH is not shown.*

In the text we currently describe 2-MGA-OS as from a high-NO pathway. In fact the formation of the initial precursor MACR is predominantly from high-NO pathways (ISOP34O2+NO, ISOPDO2+NO, nomenclature from MCMv3.3.1 (mcm.york.ac.uk)) but the formation of the direct precursor, MPAN, is from a low-NO, high $NO_2$ pathway, i.e. during the morning it would be expected that the acyl peroxy radical (MACO3) would react with NO and hence not lead to MPAN, whereas in the afternoon, with an increased $NO_2$/NO ratio, and $NO_2$ high enough to largely outcompete $HO_2$ (although 2-MGA-OS could also come from the $HO_2$ pathway via the peracid). Hence both of these isoprene oxidation products that are the precursors to the organosulfates would be expected to peak in the afternoon, as seen. In addition, the organosulfate concentrations are also dependent on the availability of particle sulphate (as seen in Bryant et al., 2020 and mentioned in the paper), which on this particular day increases over the period from 10:30 (1 $\mu g\ m^{-3}$) to 20:00 (6.5 $\mu g\ m^{-3}$). The diurnals presented for the OS

species are consistent with the chemical pathways described but are controlled by more factors than the gas phase chemistry alone. We have clarified this in the text, replacing the original lines with those below.

*"Organosulfate tracers, 2-MGA-OS (Figure 2g), and 2-methyltetrol-OS (Figure 2h), and were measured on 11 June, with low concentrations through the morning, increasing during the afternoon to a peak around 15:00-16:00. Both are tracers for low-NO chemistry, with 2-methyltetrol-OS formed via the low $HO_2$ IEPOX pathway (Paulot et al., 2009; Surratt et al., 2010; Lin et al., 2012), while 2-MGA-OS (Lin et al., 2013) is formed from the oxidation of MPAN (Kjaergaard et al., 2012; Nguyen et al., 2015), itself formed from the high-NO isoprene oxidation product MACR, in an environment with a high $NO_2/NO$ ratio, as seen in the afternoon in Beijing, and further oxidation leads to 2-MGA (Surratt et al., 2010; Chan et al., 2010; Nguyen et al., 2015)."*

ISOPONO2 concentrations may still be above 100 ppt in the afternoon, but clearly loss exceeds production in the afternoon, in contrast to the morning when production exceeds loss.

**Methods**

***Native speakers (there are a few co-authors) should help to improve especially the methods part.***

It is not particularly clear what the reviewer is referring to here. The Methods section has been reviewed by the primary authors (all native speakers), with a few minor alterations to spelling and grammar.

**CIMS**

***Discuss in more detail how the CIMS was calibrated for the isomers IEPOX and ISOPOOH, respectively. The methods description suggests that only IEPOX standards were available. Discuss measurement errors for your conditions. No calibration standards for ISOPONO2 were available. Xiong et al. 2015 found different sensitivity and stability for different ISOPONO2 isomers using I-CIMS. Discuss how this impacts the quality of your ISOPONO2 data. How are isomer specific inlet line losses estimated and corrected?***

***Xiong et al. Observation of Isoprene Hydroxynitrates in the Southeastern United States and Implications for the Fate of NOx. Atmos. Chem. Phys. 2015, 15, 11257–11272.***

As the reviewer has recognised, only IEPOX was available for calibration of the isoprene oxidation products measured with the CIMS in this study and this is already clearly stated in the text. The most analogous calibration standard to the reported measurements is therefore used here. Studies see a variation in sensitivity to different isomers and composition of course, and this will introduce small errors in the reported concentrations. Studies such as Mohr et al. (Nat. Comm. 10, 1, 2019) however report that there is a strong relationship between the sensitivity for compounds with masses > 200 Da and the collision-limit value sensitivity. Collisional limit value sensitivity was also determined in this study and there was a 14% difference in that sensitivity and the IEPOX sensitivity measured. There is an estimated maximum uncertainty of 20% in the CIMS measurements reported here, based on the variation in the suite of calibrations performed as part of this study. This is noted in the text as a limitation in terms of quantification, however in terms of the quality of the IN and

C5H10O3 time series (the most important factor for the conclusions of the paper) the data presented here is highly robust. Isomer specific line losses are not considered in the analysis of the CIMS measurements.

**PTR-MS**

*The PTR-MS instrument description suggests that ambient air containing varying ozone concentrations (low in the morning and very high in the afternoon) is sampled through a 10 L stainless steel container. Your description is not detailed enough to gain the "storage time" in this container. Describe which tests were performed to exclude/estimate production of compounds, which are monitored as 71.05 m/z ions from O3 + VOC reactions in the stainless steel container. How log is the storage time? Such artificial reactions could contribute to the observed "second" peak at 16:00-20:00 which follows the O3 concentration rather than isoprene in Fig. 2d. You mention correctly that ISOPOOH isomers are converted to MVK and MACR on stainless steel surfaces even at room temperature. If the sample air is stored in a stainless steel container for longer than seconds then the signal at mass 71.05 in Fig. 2d by no means can be assigned to MACR+MVK only! Describe what happens to ISOPOOH and IEPOX in your special inlet design.*

The PTR-MS sampled air from three locations, as described in the Methods section,

*For the first 20 minutes of each hour the PTR-MS sampled from a gradient switching manifold, and for the next 40 minutes the instrument subsampled a common flux inlet line running from the 102m platform on the tower to the container in which the PTR-ToF-MS was housed. Gradient measurements were made from 3, 15, 32, 64 and 102 m…*

The data presented in Figure 2 is the 3 m data from the gradient sampling. However, we have added the following figure to the Supplementary Information (Figure S5). This demonstrates that there is very good agreement between the MVK+MACR signal measured in the air sampled from the flux inlet line sampling at 102 m as compared to the gradient sampling at 3 m and 102 m. The flux inlet line was made of PFA tubing and had an estimated 68 s transport time from the inlet to the PTR-MS at ground level, which then directly sampled the air in contrast to the sample being drawn into stainless steel containers for the gradient sampling.

[Figure]

**Anonymous Referee #2**

**Overall:**

*This paper is novel and describes interesting field campaign results in Beijing, China that show through isoprene oxidation tracers that low-NO chemistry is important in the afternoon in Beijing, China. The paper emphasizes the importance that megacities especially those surrounded by vegetation with large biogenic VOC sources and high NOx levels likely have both low- and high-NO chemistry and modeling tools that are used for policy need to represent both of these pathways accurately.*

*Although the core of the paper is quite exciting and promising, there are gaps in the paper, which require more description before publication. The paper is well-written, but not well-organized. As described below, expanding on several sections in the main text and moving several sections from the supplement into the main text is needed before final publication.*

**In general,**

*The paper in the abstract, conclusions, and throughout, mentions the importance of accurately representing both low- and high-NO chemistry. Most models do represent both of these processes reasonably well at least for isoprene itself. Perhaps, you can expand on what you mean by this further. It may be more important to acknowledge the importance of mixed regimes. The conditions that occur in Beijing in the late afternoon are similar to the regime that occurs in the Southeast U.S. in the afternoon too. In the southeast U.S., isoprene hydroxy nitrates (formed from RO2 + NO channel) react with OH to form peroxy radicals that then react with HO2 to form isoprene dihydroxy hydroperoxy nitrates. These products formed from both RO2 + NO and RO2 + HO2 channels are significantly less studied, but important in regimes like you describe here. See references such as :*

*Xiong et al., 2015, Observation of isoprene hydroxynitrates in the southeastern United States and implications for the fate of NOx*

*Lee et al., 2015, Highly functionalized organic nitrates in the southeast United States: Contribution to secondary organic aerosol and reactive nitrogen budgets.*

> We agree with the reviewer that the chemical schemes in most regional / global models will represent both high- and low-NO chemistry to some degree, and our statements in the abstract and conclusions were not clear. We have tried now to highlight that it is the inability of models to capture the extreme diurnal cycle of NO observed that will limit the model's ability to correctly predict in-situ ozone production, SOA, etc., even though the chemical scheme within the model may be capable of representing both high and low-NO chemistry.

> We agree with the reviewer that the interplay between the high NO and low NO chemical regimes observed during the day can be expected to lead to multifunctional species of which very little is currently known with regards to their atmospheric chemistry.

*The format of the paper makes the paper harder to follow and is not similar to what is typically done in ACP. The methods are at the end of the paper instead of in the middle. The methods should be moved after the introduction and before the results.*

As discussed above, the manuscript has been formatted as a Letter (see note on formatting in the general response above), unfortunately we were not able to explicitly state this during manuscript submission.

**The results section should reference the methods section as needed.**

We agree and have now tried to include reference to the methods section where appropriate.

**Additionally, substantial information is contained in the supplement with only 3 figures in the main text. Redistributing some of the more pertinent information and figures that are currently in the supplement into the main text would be beneficial. In particular and as described below, the section on GEOS-Chem modeling with Figure S10 and the section describing how you filtered the mean diurnal plots should definitely be in the main text.**

Again this is because we have submitted the manuscript formatted as a Letter. We agree that the GEOS-Chem modelling is important to the message of the paper and this has now been included in the main text – see the specific comment below for further details. We have also included the following sentences on the filtering in the main text and directed the reader to the Supplementary Information for further details.

*"Data is filtered to only include 'typical' chemistry days, these are considered to be when ozone mixing ratios increase through the morning to an afternoon peak of > 70 ppb. Such 'typical' days account for 25 of the total of 34 measurement days. Further details of the data filtering is given in Section S1 of the Supplementary Information."*

**Additionally, references to the supplement just state "see Supplementary Information". For information that remains in the supplement after addressing the comment above, please provide more detail either by adding section numbers in the supplement or referring to specific supplementary figures.**

We have given the Supplementary Information a clearer structure, a contents page, and now refer to specific sections and figures when referenced in the main text.

**Specific comments**

**Introduction**

**There are many past papers describing atmospheric chemistry and air pollution in China and Beijing. Some summary of these studies and how this study is similar or different is warranted to put this paper in context. For example, Wang, 2017, Ozone pollution in China: A review of concentrations, meteorological influences, chemical precursors, and effects summarizes many studies.**

We have now included a paragraph on recent $NO_x$ trends in cities worldwide, with a focus on China and Beijing (given below). And also a paragraph on the source of high ozone episodes in Beijing. However, we re-iterate the point that the manuscript is not about the general atmospheric chemistry of Beijing, nor is it about photochemical ozone formation. It is about the diurnal cycle of changing oxidation pathways in Beijing. As such we do not feel that a general background of the VOC mix and atmospheric chemistry of Beijing is needed here.

*"In the past twenty years, emissions, and hence atmospheric concentrations, of nitrogen oxides ($NO_x$) have decreased in urban areas throughout the majority of the developed world. In urban areas this has been due to improvements in vehicle emissions technologies, changes to residential heating, and in many major European cities, due to restrictions on the types of vehicles that are allowed in certain areas at certain times of the day. In China, through the introduction of the "Air Pollution Prevention and Control Action Plan" in 2013 (Zhang et al. 2019) there has been a concerted effort to reduce pollutant emissions. Numerous pollution control measures have been introduced, including improved industrial emissions standards, the promotion of clean fuels instead of coal within the residential sector, improving vehicle emissions standards and taking older vehicles off the road. In Beijing, 900,000 households have converted from using coal to cleaner technologies such as gas or electricity since 2013. These actions have led to a 32 % decrease in $NO_2$ emissions since 2012 (Liu et al., 2016; Krotkov et al., 2016; Miyazaki et al., 2017). Most significant for $NO_x$ emissions however is the stringent vehicle control measures introduced within the last decade, accounting for 47 % of the total reduction in emissions for the city (Cheng et al. 2019). Such reductions in $NO_x$ emissions are expected to lead to an increased importance of low-NO oxidation pathways for VOCs in urban and suburban areas (e.g. Praske et al., 2018). This will lead to the production of a range of low volatility multi-functionalised products, efficient at producing SOA, which have previously been found only in remote environments removed from anthropogenic influence.*

*Surface ozone in Beijing has increased through the 1990s and 2000s (Tang et al., 2009). The city regularly experiences daily peaks in the summer-time of over 100 ppb (e.g. Wang et al., 2015). Such high ozone episodes are a function both of chemistry and meteorology, with air masses coming from the mountainous regions to the northwest tending to bring in clean air low in ozone, while air masses coming from the densely populated regions to the south and west bring processed polluted air high in ozone (Wang et al., 2017). A number of modelling studies have concluded that the sources of the ozone during high ozone episodes are a combination of both local production and regional transport (Wang et al., 2017; Liu et al., 2019)."*

**Page 2 line 66**

***Please include a reference here that describes the VOC mixture in Beijing. Additionally, you could add the VOC mixture at your sampling location here. How important is isoprene compared to the other VOCs you measure with the DC-GC-FID? How does your VOC mixture compare to other VOC mixtures measured in Beijing or comparable cities in China?***

Again, we do not feel that this information is pertinent to this manuscript. Isoprene oxidation products are used here as tracers. The changing diurnal oxidation pathways that are described are relevant to all VOCs. We have highlighted this point in the Discussion with the following comment:

*"With the rates of $RO_2+NO$, and $RO_2+HO_2$ similar for most peroxy radicals (Orlando and Tyndall, 2012), the chemical regime reported herein is not just relevant to isoprene, but to all*

*VOCs (see a comparison for butane and toluene in the Supplementary Information Figure S10)."*

**Figure 2**

*It would be clearer to the reader to overlay b, c, and d on top of one another and expand these figures horizontally to that of figure a. Also to overlay e and f and also expand this one horizontally to figure a. This way the NO concentration and the region of low NO chemistry can be directly seen in all of the figures.*

*Adding OH, HO2, and NO2 concentration to Figure 2 panel would be beneficial to highlight the low NO/NO2 ratio mentioned throughout the text and demonstrate how OH and HO2 change diurnally.*

> The recommended alterations to Figure 2 have been made. We have aligned all of the plots, and added the NO / $NO_2$ ratio. OH and $HO_2$ display diurnal cycles peaking in the middle of the day from 12:00-16:00 as expected and can be found in Bryant et al., 2019 (doi: 10.5194/acp-2019-929).

*The Section on "filtering data for mean diurnal plots" in the supplement should either be moved to the main text or summarized in the main text as well as in the Figure 2 caption. Without reading the supplement, the reader would not know that any filtering was done to the data. If possible, please also provide the mean diurnal profile for the same species listed in Figure 2 for the "atypical" days. This way one could contrast how the chemistry differs between a "typical" day where ozone gets above 70 ppb in the afternoon and an "atypical" day, which still occurs 25% of the time, where ozone is lower than 70 ppb. Also provide some discussion on how these products differ on "typical" and "atypical" days.*

> We have now included the following sentence in the main text, and have highlighted this in the Figure 2 caption:
>
> *"Data is filtered to only include 'typical' chemistry days, these are considered to be when ozone mixing ratios increase through the morning to an afternoon peak of > 70 ppb. Such 'typical' days account for 25 of the total of 34 measurement days. Further details of the filtering is given in Section S1 of the Supplementary Information."*
>
> A mean diurnal of the 'atypical' chemistry days would be misleading as they are not all similar, as shown for ozone in Figure S1.
>
> Unfortunately the CIMS data set is shorter than for NO and $O_3$ (2 June – 18 June), and so only contains three of the 'atypical' days. We now show NO, $O_3$, ISOPONO2, and IEPOX+ISOPOOH in Figure S1 for the two atypical days 6 June and 10 June, compared to the diurnals of the 'typical' chemistry days.

**Page 3 line 84**

*Please include references for these organosulfate tracers. Also you mention 2-MGA- OS is a tracer for high-NO SOA. Technically it is more of a tracer for high-NO and high-NO2 SOA. On page 3 line 93, you state that 2-MGA-OS is suggestive of both high and low NO chemistry. Please make sure these statements are consistent and include appropriate references.*

We agree that the description of 2-MGA as a high-NO product on line 84 is somewhat misleading. We have altered the text accordingly as described in the response to reviewer#1 above, highlighting that 2-MGA-OS requires both high-NO chemistry to form significant amounts of MACR (in the morning), and high-NO$_2$/low-NO chemistry to form MPAN (into the afternoon). References have been included for the formation of both OS species in the text along with references for the formation of the precursors.

**Page 3 line 91**

*HOMs are mentioned to have been detected at this site. There are potentially products from mixed regimes that could be detected in the late afternoon. For example, other studies have detected isoprene nitrooxy hydroxy hydroperoxides in the late afternoon when NO concentrations drop in the Southeast U.S. (see explanation above in general comments). Can you detect isoprene nitrooxy hydroxy hydroperoxides with your CIMS instrument? It would add a lot to your paper to add this compound to Figure 2 if you can detect it. This would also help to address the general comment, that it's not only about getting low- and high- NO chemistry correct, but representing products from mixed regimes that are not always incorporated into reduced chemical mechanism used in global and regional models.*

We thank the reviewers for this suggestion and based on the D'Ambro et al. (ACP., 17, 159, 2017) work from SOAS, the iodide CIMS is sensitive to species such as the groups suggested. The authors here have looked for isoprene nitrooxy **di**hydroxy hydroperoxide, C5H11NO7, formed by first addition of OH to isoprene and reaction of the peroxy radical with NO, then later addition of OH to the remaining double bond and reaction of that peroxy radical with HO2 in response to this. There are however, in the CIMS data set here, overlapping masses that given the resolving power of the instrument and the < 45 ppm difference between these identified masses it is not possible to confidently report such a measurement requested here. Identifying such compounds is clearly something for future work to focus on, in light of the findings discussed here.

*Figure3: It would be beneficial to the reader to add the year for all the campaigns (not just for NYC) as the fno will greatly depend on when the measurements were taken and how much NOx was present.*

These labels have been added.

*Page 4 line 115: The fact that GEOS-Chem cannot represent this chemistry is important and should not be hidden in the supplement. I recommend moving the description of the GOES-Chem modeling and Figure S10 into the main text or providing significantly more detail here.*

The reviewer makes a very good point, the GEOS-Chem modelling really highlights the fact that such models are unable to capture the observed diurnal for NO, and hence will get the oxidation pathways, and hence products, wrong even though the chemical schemes include both the high and low NO chemistry. We have moved all of the GEOS-Chem work from the Supplement, creating a new section in the main text, Section 4, a new section in the Methods, and making Figure S10, Figure 4.

*GEOS-chem is a global model, so please rephrase "regional chemical transport model" to include a global model nested over China or some such explanation.*

This has been changed.

**Figure S10 needs labels for which red or black lines are model versus observations.**

These lines have been added to the figure as suggested.

*There is recent evidence to suggest that aerosols play an important role in accurately representing ozone in China through loss process of NOx and HOx (Li et al., 2019). In your GEOS-chem simulations, do you assume the same aerosol uptake coefficients as this study, and if not could this impact your results in Figure S10?*

The same uptake coefficient was used as in the Li et al. (2019) study.

*Comparing more compounds (OH, HO2, isoprene, other VOCs measured by the GC-FID, ISOPOOH + IEPOX, isoprene hydroxy nitrates) with the GEOS-chem results would make your paper much more significant and help better understand how well models are representing the chemistry you describe in Beijing. Can you add this?*

A broader discussion on radical budgets in GEOS-Chem over Beijing will be the subject of a forthcoming publication. The point of the modelling here is just to show that nested global models cannot recreate the observed diurnal cycle of NO in Beijing, which appears to be caused by missing processes in our chemical understanding rather than problems with emissions inventories.

*If aerosols are important in China for representing O3-NOx-HOx (and in particular the loss of HO2 to aerosols), how would this impact your results with the box-modeling in Figure 2, which only includes gas-phase chemistry.*

This is of course a complex issue, as we mention in the *Discussion and Conclusions*. On the face of it, $HO_2$ reductions caused by uptake to aerosol might be expected to reduce $f_{HO2}$, and hence increase $f_{NO}$. However, additional feedbacks would also occur, such as the increased ozone associated with reduced $HO_2$ (Li et al., 2019), which would suppress [NO] and bring the $f_{NO}/f_{HO2}$ ratio back the other way. In short, heterogeneous uptake of $HO_2$ may be important in certain urban environments, but this importance will be variable on a daily and seasonal basis. The box modelling is a simple representation of the competition between NO and $HO_2$ for reaction with peroxy radicals, and while an additional $HO_2$ sink may shift the plot slightly (though maybe not due to the feedbacks mentioned above), it would not change the overall take home message of the plot. Moreover, the recently published experimental study of Tan et al. (2020, ES&T, doi: 10.1021/acs.est.0c00525) conducted in the North China Plain in the summer of 2014 – observed insignificant effects of heterogeneous uptake of $HO_2$ to aerosol on the radical budget, and hence on ozone formation, in contrast to the modelling of Li et al. (2019).

*On page 5 line 137, Jacob et al., 2019 is referenced, but not incorporated in the reference list. Please update this reference or use Li et al., 2019.*

This has been corrected in the text to Li et al. (2019).

**Page 9 line 274:** *What was the relative humidity used in the box-model?*

A fixed relative humidity of $0.01 * N_A$ was used, i.e. $2.55 \times 10^{17}$ molecules $cm^3$. This is now mentioned in the *Box Modelling* section of the Methods.

---

## Author Response (AR2)

**Response to Reviewers**

**26/10/20**

**Response to Reviewers of 1st revision of:**

**Rainforest-like Atmospheric Oxidation Pathways in a Polluted Megacity by Newland et al., 2020, submitted to ACP**

**General Response**

We thank the editor and reviewers for giving up more of their time to make further insightful comments, helping to clarify and further improve our manuscript.  The editor and reviewers recognise the importance of the results
presented, and recommend publication in ACP after some minor changes. All changes to the manuscript are in line with the editor's and reviewers' comments and suggestions.

Responses to the editor and to each reviewer are given below. Responses to specific points raised by each reviewer are given separately beneath that point. Reviewers' comments are bold and italic, the authors'
comments are inset in plain type.

**Editor's Comments**

**General Comments**

*1. Like reviewer 2, I am not convinced that these are "rainforest-like" conditions. The work describes a missing sink of NO that does not appear to produce ozone (similar to the Hofzumahaus publication referenced in the manuscript and as you point out). This means that the low NO has an unknown cause. This unknown chemistry puts in question whether the chemical regime is the same as in rainforests, or are the authors implying that the same unknown chemistry to keep NO low is active there. Figure 3 also is not very convincing for the conditions*
*in Beijing from this work to be similar to those of Borneo and the rural SE-USA, although they are clearly different than London or NY. I think a clarification would be useful.*

Since this still appears to be confusing matters, we are happy to modify the title further to, '***Low NO***
***Atmospheric Oxidation Pathways in a Polluted Megacity'.*** We think that subsequent references to
'rainforest-like' in the text are hopefully acceptable, and when mentioned that it is clear that we are referring to either the fact that there is low NO, or that the products observed are the same as those observed in VOC oxidation in a rainforest. E.g. we clarify this in the abstract saying that '… we observe significant formation of gas and aerosol phase oxidation products associated with the low-NO 'rainforest-like' regime…', highlighting that the reference to 'rainforest-like' refers to the mixing ratio of NO. We are
not implying that the reason for the low NO is similar to the reason for low NO in a rainforest.

*2. Unless I missed it, the manuscript does not comment on uncertainty or variability of measurements, with the exception of figure 4. Therefore, it is not possible to draw conclusions on whether there are statistically significant differences. Calibrations are only mentioned very briefly. For example, in figure S5 that one of the reviewer 2 comments one, it is not clear whether there are statistically significant differences or not. Uncertainty and variability should be considered in comparisons.*

We have created a version of Figure 2 that shows the standard deviation of the mean for the diurnals. (These are not shown for the particle phase measurements as they are only from one day). We think that, while it is of course useful to show the variability in the data, adding the uncertainty makes the diurnal trends shown in Figure 2 more difficult to discern due to the necessity of extending the y-axis. Therefore, we have placed the figure with the standard deviations shown in the Supplementary Information as Figure S3, with a note added at the end of the first sentence in the caption of Figure 2 (below). If the editor considers that these should be shown on Figure 2 in the main manuscript then we could do this but may have to redesign the figure a little, changing the dimensions to make it higher and less wide.

*Mean diurnal variation of measured organic and inorganic species in the gas phase and aerosol during the Beijing summer observations (data is filtered to only include 'typical' chemistry days – see text for details; the standard deviation of the mean is shown in Figure S3).*

**CIMS:** The calibration uncertainty on the CIMS measurements of IEPOX+ISOPOOH and ISOPONO2 is estimated at 50% based on the calibration using a standard for IEPOX. A further sentence to this effect has been added at the end of the CIMS part of the Methods section:

*Absolute measurement uncertainties are estimated at 50% for the presented IEPOX+ISOPOOH and ISOPONO2 ($C_5H_9NO_4$) signals.*

**PTR:** The calibration uncertainty on the PTR measurements of m/z 71.05 has been estimated to be 21% using a transmission curve (Taipale et al., 2008). A further sentence to this effect has been added at the end of the PTR part of the Methods section:

*The sensitivity for each mass was then calculated using a transmission curve. The maximum relative error for PTR-MS calibration using a relative transmission curve has been estimated to be 21% (Taipale et al., 2008).*

*3. Similarly, the abstract has the statement "significant formation of gas and aerosol phase oxidation products associated with the low-NO 'rainforest like' regime". The word significance should be replaced with a quantitative number as it is not clear what significant means here. Are the amounts of ISOPOOH and IEPOX that are observed significant and what are they significant for? Similarly, are 20 ng/m3 of 2-MT-OS significant? The manuscript does not discuss how much overall SOA was observed and how much of it was from low-NO pathways, so the statement that there is significant formation of aerosol phase oxidation products is hard to evaluate.*

The significance of the formation of the low NO products is that they demonstrate that low NO pathways are active. However, we acknowledge that we are not using the word *significant* here in a numerical sense and so have removed it. The significance in a more numerical sense is given a couple of sentences later in the abstract in the numbers from the modelling, i.e. that up to ~ 30 % of $RO_2$ are following low NO pathways in the afternoon.

The mean concentration of 2-MT-OS (11.8 ng $m^{-3}$) is of course lower than observed in the Amazon and SE US, but reaches up to a maximum of 100 ng $m^{-3}$ or 1.2 % of the OOA measured by AMS (Bryant et al.,2020). In previous campaigns, mean 2-MT-OS mass concentrations were 83(wet)/399(dry) ng $m^{-3}$ in the Amazon (Glasius et al., 2018), 169.5 ng $m^{-3}$ in the SE USA (Budisulistiorini et al., 2015) and 5.3 ng $m^{-3}$ in rural China, near Beijing (He et al., 2018). There is a lack of quantified data from other cities for comparison and so future work is needed to compare the importance of low NO oxidation pathways in cities where NO mixing ratios are higher.

In addition, recent studies indicate that 2-MT-OS may undergo further heterogeneous oxidation in highly oxidative environments such as Beijing, and so this is likely to be a lower limit for the amount that has actually formed through this pathway.

In this study we are using the IEPOX and 2-MT-OS data to show that low NO pathways can form SOA in Beijing. This is further evidenced by the observation of HOMs in the afternoon in Beijing by Brean et al. (2019) – already referenced in the manuscript. Furthermore, positive matrix factorisation of FIGAERO-CIMS data during the same campaign produced a factor that peaked in the afternoon (PM-OOA2) and had a large contribution from HOMs and isoprene oxidation products (Mehra et al. 2020; Faraday Discuss.). However, at this point much more research is needed to be able to determine the % of SOA forming from low NO chemistry.

Bryant, D. J., Dixon, W. J., Hopkins, J. R., Dunmore, R. E., Pereira, K. L., Shaw, M., Squires, F. A., Bannan, T. J., Mehra, A., Worrall, S. D., Bacak, A., Coe, H., Percival, C. J., Whalley, L. K., Heard, D. E., Slater, E. J., Ouyang, B., Cui, T., Surratt, J. D., Liu, D., Shi, Z., Harrison, R., Sun, Y., Xu, W., Lewis, A. C., Lee, J. D., Rickard, A. R., and Hamilton, J. F.: Strong anthropogenic control of secondary organic aerosol formation from isoprene in Beijing, Atmos. Chem. Phys., 20, 7531–7552, https://doi.org/10.5194/acp-20-7531-2020, 2020.

Glasius, M., Bering, M. S., Yee, L. D., De Sá, S. S., Isaacman-VanWertz, G., Wernis, R. A., Barbosa, H. M. J., Alexander, M. L., Palm, B. B., Hu, W., Campuzano-Jost, P., Day, D. A., Jimenez, J. L., Shrivastava, M., Martin, S. T., and Goldstein, A. H.: Organosulfates in aerosols downwind of an urban region in central Amazon, Environ. Sci. Process. Impact., 20, 1546–1558, https://doi.org/10.1039/c8em00413g, 2018.

He, Q. -F., Ding, X., Fu, X. -X., Zhang, Y. -Q., Wang, J. -Q., Liu, Y. -X., Tang, M. -J., Wang, X. -M., and Rudich, Y.: Secondary Organic Aerosol Formation From Isoprene Epoxides in the Pearl River Delta, South China: IEPOX- and HMML- Derived Tracers, J. Geophys. Res.-Atmos., 123, 6999–7012, https://doi.org/10.1029/2017JD028242, 2018.

Budisulistiorini, S. H., Li, X., Bairai, S. T., Renfro, J., Liu, Y., Liu, Y. J., McKinney, K. A., Martin, S. T., McNeill, V. F., Pye, H. O. T., Nenes, A., Neff, M. E., Stone, E. A., Mueller, S., Knote, C., Shaw, S. L., Zhang, Z., Gold, A., and Surratt, J. D.: Examining the effects of anthropogenic emissions on isoprene- derived secondary organic aerosol formation during the 2013 Southern Oxidant and Aerosol Study (SOAS) at the Look Rock, Tennessee ground site, Atmos. Chem. Phys., 15, 8871–8888, https://doi.org/10.5194/acp-15-8871-2015, 2015.

*4. According to Nguyen et al. (PCCP 17, 17914-17926, 2015) MPAN produces SOA via oxidation by OH to form HMML without any participation NO or HO2, so it would be useful if the authors could explain how SOA from MPAN indicates low NO conditions.*

The formation of 2-MGA-OS requires high and then low NO conditions (in fact not necessarily low NO but a low NO/NO$_2$ ratio). MACR is formed (predominantly) from oxidation of isoprene in a high NO environment. Then the afternoon low NO/NO$_2$ ratio leads to efficient formation of MPAN from OH initiated MACR oxidation and subsequent oxidation to 2-MGA via HMML.

We have clarified this in the text changing it from:

*Both are tracers for low-NO chemistry, with 2-methyltetrol-OS formed via the low HO2 IEPOX pathway (Paulot et al., 2009; Surratt et al., 2010; Lin et al., 2012), while 2-MGA-OS (Lin et al., 2013) is formed from the oxidation of MPAN (Kjaergaard et al., 2012; Nguyen et al., 2015), itself formed from the high-NO isoprene oxidation product MACR, in an environment with a high NO2/NO ratio, as seen in the afternoon in Beijing, and further oxidation leads to 2-MGA (Surratt et al., 2010; Chan et al., 2010; Nguyen et al., 2015).*

To:

*Both are tracers for non-NO RO$_2$ chemistry. While 2-methyltetrol-OS is formed via the ISOPOO+HO$_2$ IEPOX pathway (Paulot et al., 2009; Surratt et al., 2010; Lin et al., 2012), 2-MGA-OS (Lin et al., 2013) is formed from the OH initiated oxidation of MPAN (Kjaergaard et al., 2012; Nguyen et al., 2015), with further oxidation leading to 2-MGA (Surratt et al., 2010; Chan et al., 2010; Nguyen et al., 2015). MPAN is a product of the OH initiated oxidation of MACR in an environment with a high NO$_2$/NO ratio. So the observation of 2-MGA-OS formation reflects the observed diurnal NO cycle in Beijing. MACR is formed in the morning through the OH oxidation of isoprene in a high NO environment, followed by OH oxidation of MACR in a high NO$_2$/NO environment in the afternoon to form MPAN, which reacts further with OH to yield 2-MGA.*

*5. Lastly, it would be useful to comment on HO2 measurements, which LIF FAGE instruments often measure, as a comparison could help evaluate the fNO number as it in large part depends on competition between NO and HO2.*

The HO$_2$ measurements from the campaign are now available in Whalley et al. (2020) (doi: 10.5194/acp-2020-785). The mean diurnal [HO$_2$] for the campaign peaks in the mid-afternoon with a campaign mean for the daily peak of roughly $5 \times 10^8$ cm$^{-3}$, although on some days this is as high as $8 \times 10^8$ cm$^{-3}$. With NO falling to < 1 ppb on most days in the afternoon, and sometimes to < 100 ppt, the calculated contribution of the low NO pathway is in the range of 10 – 50 %, as borne out by the model. We have included the following in the discussion in Section 3: Box Modelling.

*$HO_2$ concentrations were measured by FAGE during the campaign (Whalley et al., 2020). Concentrations peaked in mid-afternoon (i.e. when NO is at its lowest), regularly exceeding $5 \times 10^8$ cm$^{-3}$ and reaching up to $1 \times 10^9$ cm$^{-3}$ on some days. Based on the relative reaction rates of $RO_2$ with NO and $HO_2$, for $[HO_2] = 5 \times 10^8$ cm$^{-3}$ the contribution of low NO pathways to $RO_2$ removal would be expected to be roughly 50 % at [NO] = 100 ppt and 10 % at [NO] = 1 ppb.*

Box modeling using the MCM and constraining to the measurements from Beijing in Whalley et al. (2020) (similar to that done in this paper) tends to over-predict the $HO_2$ concentrations measured by the FAGE. One solution put forward by Whalley et al. that can greatly improve model-measurement comparison of $[HO_2]$ is the inclusion of some parameterised HOMs / auto-oxidation chemistry, in which $RO_2$+NO leads to further $RO_2$ rather than to $HO_2$. This effectively increases the number of $RO_2$+NO reactions per $HO_2$ formed. This would also qualitatively fit with our observation of a missing NO sink. It is now well recognised that the current generation of semi-explicit chemical mechanisms (e.g. MCMv3.3.1) are missing much of this auto-oxidation, H-shift chemistry and this is one of the main foci in developing new generations of such mechanisms. This is even more the case for the reduced mechanisms included in global models such as GEOS-Chem. We have already touched on this in the discussion as a possible explanation for the inability of GEOS-Chem to replicate the observed diurnal cycle of NO. We have now expanded this discussion slightly.

This paragraph has changed from:

*The fact that the GEOS-Chem modelling cannot recreate the extremely low afternoon NO suggests that there may be additional sinks for NO beyond our current chemical understanding. One explanation may be additional NO sinks that recycle OH without producing $O_3$, as previously proposed for the high VOC-low NO (< 1ppbv) conditions seen in Beijing and other cities*

*(Hofzumahaus et al., 2009; Whalley et al., 2018; Tan et al., 2019). Another explanation may be the presence of high concentrations of other species that can rapidly convert NO to $NO_2$ e.g. halogen oxides.*

To:

*The fact that the GEOS-Chem modelling cannot recreate the extremely low NO levels observed in the afternoon suggests that there may be additional sinks for NO beyond those currently included in the chemical mechanism. One explanation may be the occurrence of $RO_2$+NO oxidation pathways that lead to the formation of a second $RO_2$ before forming a stable species, effectively*

*increasing the efficiency of NO to $NO_2$ conversion per initial oxidation step (e.g. Whalley et al., 2020). Such reactions are expected to be particularly important for larger and more complex VOCs, for which the detailed oxidation processes have been less studied and which are heavily parameterised in global models. Auto-oxidation processes that regenerate OH, leading to the formation of further $RO_2$ have also been proposed previously for the high VOC-low NO (< 1ppbv)*

*conditions seen in Beijing and other cities (Hofzumahaus et al., 2009; Whalley et al., 2018; Tan et al., 2019). Inclusion of such novel $RO_2$ oxidation processes is one of the main foci of the next generation of atmospheric chemical mechanisms. Another explanation may be the presence of*

*high concentrations of other species (not currently included in the chemical scheme) that can rapidly convert NO to NO$_2$, e.g. halogen oxides.*

Whalley, L. K., Slater, E. J., Woodward-Massey, R., Ye, C., Lee, J. D., Squires, F., Hopkins, J. R., Dunmore, R. E., Shaw, M., Hamilton, J. F., Lewis, A. C., Mehra, A., Worrall, S. D., Bacak, A., Bannan, T. J., Coe, H., Ouyang, B., Jones, R. L., Crilley, L. R., Kramer, L. J., Bloss, W. J., Vu, T., Kotthaus, S., Grimmond, S., Sun, Y., Xu, W., Yue, S., Ren, L., Acton, W. J. F., Hewitt, C. N., Wang, X., Fu, P., and Heard, D. E.: Evaluating the sensitivity of radical chemistry and ozone formation to ambient VOCs and NO$_x$ in Beijing, Atmos. Chem. Phys. Discuss., https://doi.org/10.5194/acp-2020-785, in review, 2020.

**Anonymous Referee #1**

*The aim of the revised manuscript by Newland et al. is now tightly focused on the key, high impact result: The authors use isoprene oxidation products as photochemical markers of the changing chemical pathways throughout the daytime and demonstrate that the model used to capture this diurnal change fail.*

*Figure 4 shows that measured NO is high in the morning and low in the afternoon.*
*It is not surprising that isoprene oxidation forms typically high NO products in the morning and low NO products in the afternoon.*

We agree that, on the face of it this is not surprising, but this is exactly the point of this Letter, the suite of gas and aerosol phase product observations corroborate the observed diurnal suppression of NO and
highlight the effects of this chemical cycle – i.e. the production of OVOC / SOA products that are associated with low-NO chemistry and are not typically expected to be found in a highly polluted urban environment.

*At low NO, RO2 radicals especially ISOPOO are reacting with HO2 and RO2 rather than with NO. In Beijing the*
*complex VOC composition leads to a complex mixture of RO2 radicals. The reaction between these RO2 radicals might be the key to get a better match with measurements when included in models.*

Yes, we agree that this is what is happening. Although $RO_2 + RO_2$ reactions are likely to be negligible since $[RO_2]$ are likely to be lower than $[HO_2]$ and $k_{HO2+RO2}$ is generally at least two orders of magnitude greater
than $k_{RO2+RO2}$ (though there may be some surprisingly fast $RO_2$ cross reaction rates - and this is certainly an area that requires further research at a laboratory kinetics level).

*The authors explicitly deny to describe the VOC composition in Beijing, talk about the role of isoprene chemistry in Beijing, or talk about ozone production in Beijing.*
*Therefore I question the key, high impact of the result presented here.*

As highlighted in the previous response, this discussion is out of scope of this highly focused Letter, but is important and the subject of other publications led by some of the co-authors (which have in the
meantime been published). Further details of these points are now available in Whalley et al. (2020) (doi: 10.5194/acp-2020-785). We maintain that, while these are undoubtedly of interest scientifically, they are not relevant to the message of this Letter.

*As explained in the next paragraph measured PTR-MS signal at 71.05 m/z cannot be assigned quantitatively to MVK+MACR. The new Figure S5, thought to support MVK+MACR assignment of signal at 71.05 m/z, is misinterpreted.*

*Fig. S5 demonstrates that there is agreement between the PTR-MS signal at 71.05 m/z measured in the morning from 7:00 to 11:00 comparing different inlet lines. But there is a striking difference in the afternoon. From 16:00 to 18:00 both lines taking air at 102 m show 0.8 ppb (flux line, PFA, transport time 68 s) vs. < 0.6 ppb (gradient line, sample being drawn into stainless steel containers).*

> We think that the reviewer has misinterpreted which line refers to which sampling method in Figure S5. The deviation at 17:00 is between the two gradient sampling lines and the flux line (rather than the two methods sampling from 102 m and that from 3 m as suggested by the reviewer).

*In the morning when high NO isoprene chemistry produces prevailingly MVK + MACR that is detected at 71.05 m/z the agreement between the two inlet lines from the same intake point is reasonable. In the afternoon under low NO, MVK, MACR, ISOPOOH and IEPOX is formed contributing with different sensitivities to the 71.05 m/z PTR signal.*

*All PTR-MS data and discussion termed MVK+MACR have to be taken out from the manuscript.*

> See comment on Figure 2f below. The line where MACR+MVK is first mentioned has been changed from:

>> *The profile of the high-NO products MACR+MVK, measured by Proton Transfer Reaction Time-of-Flight Mass Spectrometer (PTR-ToF-MS) – see Methods for further details, is very similar to that of ISOPONO$_2$ until about 15:00, when they begin to increase, with a second peak observed at around 17:00 (Figure 2f).*

> To:

>> *The profile of the signal at m/z 71.05 is assumed to be dominated by the high-NO products MACR+MVK, measured by Proton Transfer Reaction Time-of-Flight Mass Spectrometer (PTR-ToF-MS) – see Methods for further details. This signal is very similar to that of ISOPONO$_2$ until about 15:00, when it begins to increase, with a second peak observed at around 17:00 (Figure 2f).*

*Line 587: The PTR-TOF-MS signal at 71.05 m/z is not MVK+MACR. Fig. S5 demonstrates that different inlet lines lead to strongly different signal intensities erroneously termed MVK+MACR.*

> It is unclear how the reviewer interprets Figure S5 as demonstrating strongly different signal intensities.

*Line 590: 1,2-ISOPOOH, which is mainly produced in isoprene OH reactions converts at stainless steel surfaces to MVK.*

> The reviewer makes a good point, this sentence has been changed from:

>> *This latter peak may be an observational artefact as a result of the conversion of ISOPOOH to MACR on metal surfaces within the inlet of the PTR instrument (Rivera-Rios et al., 2014).*

To:

*This latter peak may be an observational artefact as a result of the conversion of ISOPOOH to either MVK (via 1,2-ISOPOOH - the dominant isomer (Reeves et al., 2020)) or MACR (via 4,3-ISOPOOH) on metal surfaces within the inlet of the PTR instrument (Rivera-Rios et al., 2014).*

Reeves, C. E., et al.: Observations of speciated isoprene nitrates in Beijing: implications for isoprene chemistry, Atmos. Chem. Phys. Discuss., https://doi.org/10.5194/acp-2019-964, in review, 2020.

**Figure 2f: The assignment of the signal at 71.05 m/z as MVK+MACR is not correct. Figure 2f has to be taken out.**

Figure 2f has not been taken out, but the axis has been changed to m/z 71.05 rather than MACR+MVK. Thus readers can see in the text and the figure caption the discussion of the attribution of m/z 71.05 to MACR+MVK or potentially to other sources such as ISOPOOH. The following has been added to the figure caption:

*(f) m/z 71.05, assumed to be predominantly the gas phase isoprene 'high NO' oxidation products methacrolein (MACR) (precursor to 2-MGA) + methyl vinyl ketone (MVK), signal calibrated with MACR/MVK – see text for further details.*

**The new title "Rainforest-like Atmospheric Oxidation Pathways in a Polluted Megacity" is misleading because: The key result here is connected with changing chemical pathways throughout the daytime due to NO availability. There is not such a NO change in the rainforest! The VOC composition and reactivity is also very much different in the rainforest. Therefore I suggest to take out the word "rainforest" from the title, which is misleading.**

The word rainforest has now been removed from the title – see reply to the editor's comment for further detail.

**Anonymous Referee #2**

**The revisions made in the previous review stage are sufficient for publication with one minor update:**

**Figure 4: In the previous submission the ozone observations agreed with the model results quite well. However, in the updated version of the manuscript, ozone is not represented well by the model contrary to what is stated in the text. It is important to note that the model under-predicts ozone in the mean (particularly in the**
**afternoon) and the spread (if I interpret the shading correctly?) is also very different with the high ozone days in the observations clearly missed within the model. To make this point clearer, please specify in the figure 4 caption what the shading represents? Is this the 25th to 75th percentile? With this updated figure, it appears that the model's inability to represent the low NO values is potentially connected with not representing the high ozone events correctly and this should be noted in the text.**

The shaded areas represent +/- two standard deviations of the mean of the diurnals. This has now been noted in the figure caption.

The reviewer is correct that the model under-predicts the measured ozone diurnal peak by about 10 %
(this is a little less than in the previous version of the figure – see comments below for further details on slight changes to the way the figure has been presented). This has now been noted in the Discussion section (below). But also noted is that this slight under-prediction of ozone cannot be responsible for the overprediction of [NO], which is over-predicted in the afternoon by a factor of 3 – 5.

*The mean afternoon ozone peak is under-predicted by the model by about 10 %. However, this has little impact on the modelled NO concentrations which the model over-predicts by a factor of 3 - 5 through the afternoon.*

The GEOS-Chem modelling presented in the supplement of the original version of the manuscript was
compared to diurnals calculated using the full measurement dataset from the campaign (not the filtered data presented in Figure 2). For the revised version of the manuscript, in which the GEOS-Chem modelling became Figure 4, we realised that we were now presenting different measurement data for $NO_x$ and $O_3$ in Figure 4 compared to Figure 2 (the diurnal means of the full campaign compared to the diurnal means of the filtered data). We felt that we should present the same measurements in both plots so this now
compared the diurnals calculated using the filtered data to modelled diurnals for the whole campaign. (Ideally, we would perform the GEOS-Chem modelling with the filtered days removed. However, it is not apparently possible to simply filter out days for the GEOS-Chem modelling). This meant that the modelling was now somewhat underpredicting the ozone compared to the measurements (as low ozone days are filtered out of the measurements).

In addition, the GEOS-Chem modelling in the original manuscript scaled isoprene emissions regionally, whereas the modelling in the revised version scaled the isoprene emissions locally over the Beijing urban area. This had a small effect on modelled ozone, reducing the modelled diurnal peak by about 10 %.

We now present the modelling using the locally scaled isoprene emissions, and compare to the diurnals calculated from the full measurement dataset. This is now noted in the figure caption and in the text. A line has also been added about the scaling of isoprene emissions at the end of the GEOS-Chem modeling description in Methods.

      *Isoprene emissions calculated by the MEGAN biogenic emissions extension were scaled by 2.5x in the Beijing metropolitan region (Jing-Jin-Ji).*

None of this changes the general conclusion that the GEOS-Chem model is unable to recreate the observed diurnal NO cycle. And this is the case whether the filtered or unfiltered dataset is used.

**Marked-up
Manuscript Version**

[revised manuscript text omitted]

---

## Author Response (AR3)

5

**Response to**
**Handling Editor**

10

**02/12/20**

**Response to the handling editor of:**

**Low-NO Atmospheric Oxidation Pathways in a Polluted Megacity by Newland et al., 2020, submitted to ACP**

**General Response**

We thank the editor for giving up more of their time to make further insightful comments, helping to clarify and further improve our manuscript.

Responses to the editor are given below. Responses to specific points raised by the editor are given separately beneath that point. The editor's comments are bold and italic, the authors' comments are inset in plain type.

Note, we have not included a further marked up version of the manuscript since only one sentence has changed. And the methods has moved from the end of the manuscript to become Section 2.

**Editor's Comments**

**General Comments**

*Dear authors, thank you for your detailed response. I just have one more comment. On line 146-147 it is stated that "The observed temporal profiles of the isoprene tracer products suggest a chemical cycle switching from a high-NO to a low-NO chemical regime during the day in Beijing." To a casual reader this sounds like low-NO dominates. However, if I understand the manuscript correctly, and as stated in the abstract the low-NO accounts for ca. 30% which is not dominating. Would you consider rephrasing that as stating something like that there is a clear shift from high-NO chemical regime to an important contribution from low-NO or something similar that you are comfortable with.*

We agree with this clarification, and have changed the sentence from:

*The observed temporal profiles of the isoprene tracer products suggest a chemical cycle switching from a high-NO to a low-NO chemical regime during the day in Beijing.*

To:

*The observed temporal profiles of the isoprene tracer products suggest a chemical cycle switching from a high-NO chemical regime in the morning, to a regime with a significant contribution from low-NO chemistry in the afternoon in Beijing.*